# PYSPATIAL: GENERATING 3D VISUAL PROGRAMS FOR ZERO-SHOT SPATIAL REASONING

**Zhanpeng Luo[1,2]*   Ce Zhang[1]*   Silong Yong[1]   Cunxi Dai[1]   Qianwei Wang[1,3]**
**Haoxi Ran[1]   Guanya Shi[1]   Katia Sycara[1]   Yaqi Xie[1]**
[1]Carnegie Mellon University   [2]University of Pittsburgh   [3]University of Michigan

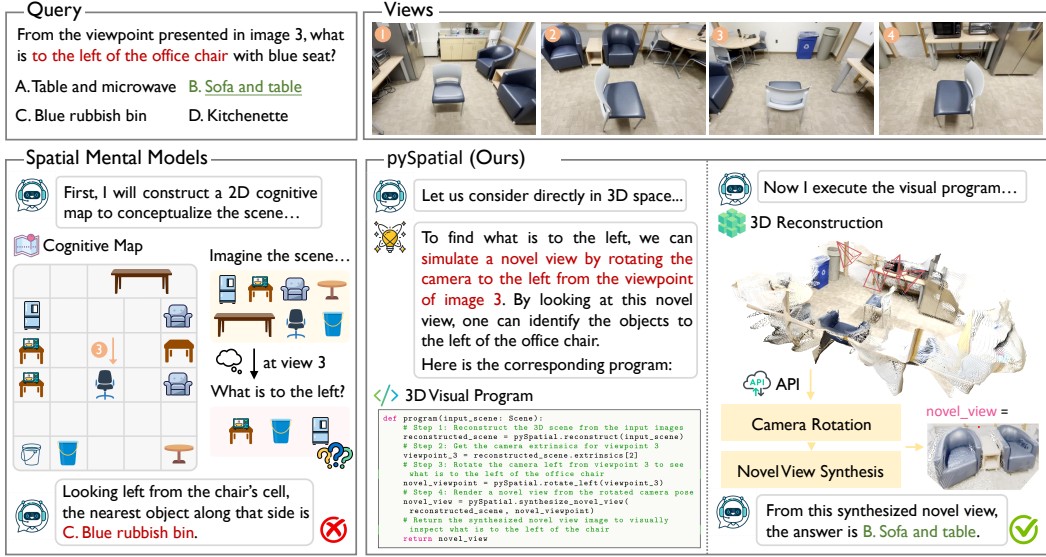

**Figure 1: Comparing our `pySpatial` with spatial mental models for multi-view spatial reasoning tasks**. Unlike spatial mental models (Yin et al., 2025), which rely on the implicit imagination of MLLMs to construct a 2D cognitive map, we introduce `pySpatial`, a visual programming framework that flexibly composes spatial tools (*e.g.*, 3D reconstruction, camera movements, and novel view synthesis) to enable MLLMs to explicitly reason in 3D space for diverse spatial reasoning tasks.

## ABSTRACT

Multi-modal Large Language Models (MLLMs) have demonstrated strong capabilities in general-purpose perception and reasoning, but they still struggle with tasks that require spatial understanding of the 3D world. To address this, we introduce `pySpatial`, a visual programming framework that equips MLLMs with the ability to interface with spatial tools via Python code generation. Given an image sequence and a natural-language query, the model composes function calls to spatial tools including 3D reconstruction, camera-pose recovery, novel-view rendering, *etc*. These operations convert raw 2D inputs into an explorable 3D scene, enabling MLLMs to reason explicitly over structured spatial representations. Notably, `pySpatial` requires no gradient-based fine-tuning and operates in a fully zero-shot setting. Experimental evaluations on the challenging MINDCUBE and OMNI3D-BENCH benchmarks demonstrate that our framework `pySpatial` consistently surpasses strong MLLM baselines; for instance, it outperforms GPT-4.1-mini by 12.94% on MINDCUBE. Furthermore, we conduct real-world indoor navigation experiments where the robot can successfully traverse complex environments using route plans generated by `pySpatial`, highlighting the practical effectiveness of our approach. Our project website is available at https://pySpatial.github.io.

---

*Equal contribution.

## 1 INTRODUCTION

Multi-modal Large Language Models (MLLMs) have achieved remarkable success across diverse tasks such as image captioning (Bucciarelli et al., 2024; Zhang et al., 2025b), referring grounding (Kazemzadeh et al., 2014), video understanding (Zeng et al., 2025; Fu et al., 2025), and robust reasoning (Mathew et al., 2021; Zhang et al., 2025a). However, this progress has not translated into robust 3D spatial reasoning: recent studies (Wu et al., 2025; Chen et al., 2024a; Chang et al., 2025) reveal that MLLMs still struggle with challenges spanning from basic tasks such as judging relative object positions or estimating depth in a single image (Liu et al., 2023; Cheng et al., 2024) to more complex reasoning over egocentric motion and multi-view relations (Yin et al., 2025; Yang et al., 2025). Such limitations pose a substantial barrier to their reliable deployment in safety-critical applications including robotics, augmented reality, and embodied intelligence, where tasks such as navigation, manipulation, and human–robot interaction depend on precise spatial understanding (Li et al., 2024; Duan et al., 2024; Song et al., 2025; Qiao et al., 2025).

While recent efforts (Chen et al., 2024a; Cheng et al., 2024) have primarily targeted improving spatial understanding from a single image (*e.g.*, "*Is the stool in front of the oven?*"), in this work we focus on the more challenging problem of 3D spatial reasoning, where the environment is only partially observed with limited views and models must reason across perspectives to answer queries such as "*Where should I move from view 1 to view 2?*"—a setting in which state-of-the-art MLLMs perform only slightly above random guess (Yin et al., 2025). Recent studies (Chen et al., 2024a; Ma et al., 2025) suggest that this weakness largely stems from the training data: although MLLMs are pre-trained on internet-scale image-caption pairs, explicit 3D supervision is sparse and costly, making it difficult to learn reliable correspondences between language and 3D spatial structures and thereby constraining models' ability to reason effectively in 3D space. More recently, Yin et al. (2025) explores the use of data structures such as 2D cognitive maps, where the model encodes object positions in a top-down view to mentally simulate spatial layouts, as shown in Figure 1. However, these approaches still rely on implicit "imagination" mechanisms and offer only limited effectiveness.

These limitations motivate our central research question: *how can we equip MLLMs with explicit reasoning capabilities in 3D space?* A natural first step toward this goal is to obtain an explicit geometric foundation on which such reasoning can take place. Recent advances in feed-forward 3D reconstruction (Wang et al., 2024b; 2025a) makes this feasible by recovering scene geometry directly from sparse 2D views, including camera parameters, depth maps, and scene-level point clouds. Such representations transform limited 2D views into an *explorable* 3D scene, within which models can perform spatial transformations (hereafter referred to as *spatial tools*) such as camera translation, rotation, and viewpoint shifts to enrich visual context and build interactive reasoning chains. For instance, given the query "*what is behind me if I am at view 3*," the model could rotate the virtual camera by 180° at the specified viewpoint within the reconstructed scene, thereby uncovering previously occluded regions and grounding its reasoning in geometric evidence.

However, how to enable MLLMs to flexibly compose spatial tools and seamlessly interact with 3D environments in a context-aware manner remains a critical challenge. To address this, inspired by pioneering works on visual programming (Gupta & Kembhavi, 2023; Surís et al., 2023), we introduce pySpatial, a framework that employs MLLMs like GPT-4o as Python code generation agents to invoke function calls for tools such as 3D reconstruction, natural language description of movements, and novel view synthesis. As illustrated in Figure 1, pySpatial leverages a well-defined API to automatically select and compose the appropriate tools to solve diverse spatial reasoning tasks. Notably, pySpatial operates fully in a zero-shot setting and serves as a plug-and-play framework applicable to both open-source and closed-source MLLMs, offering interpretable solutions and reliable responses that make it well-suited for diverse real-world tasks.

We evaluate the effectiveness of our approach on the MINDCUBE and OMNI3D-BENCH benchmarks, where results demonstrate that pySpatial consistently outperforms strong MLLM baselines by substantial margins (*e.g.*, achieving a 12.94% improvement over GPT-4.1-mini on MINDCUBE). Qualitative analyses further verify that our approach can generate high-quality executable and interpretable visual programs that can effectively solve complex spatial reasoning tasks in a zero-shot manner. Furthermore, we apply pySpatial to real-world indoor navigation, where it successfully enables a quadrupedal robot to traverse complex environments using generated route plans.

Our contributions can be summarized as follows:

- We present pySpatial, a novel zero-shot framework that enables MLLMs to reason explicitly in 3D space by generating and executing visual programs that leverage various spatial tools in a structured, compositional manner to solve diverse spatial reasoning tasks.
- We evaluate pySpatial on MINDCUBE and OMNI3D-BENCH, where it demonstrates superior performance over strong MLLM baselines. Qualitative analysis validates that pySpatial reliably generates executable and interpretable visual programs for diverse spatial reasoning tasks.
- We further assess the practical effectiveness of pySpatial on indoor navigation tasks, showing that it can generate route plans that enable a quadrupedal robot to traverse complex environments, demonstrating strong potentials for practical use cases.

## 2 RELATED WORK

**MLLMs for Spatial Reasoning**. Recent MLLMs have demonstrated remarkable performance on multi-modal tasks (Wan et al., 2025; Alayrac et al., 2022; Zhang et al., 2024; 2026). However, studies have shown that these models exhibit significant limitations in interpreting spatial relations (Yu et al., 2024; Kamath et al., 2023; Wang et al., 2024a; Tong et al., 2024), a critical precursor to a wide range of practical applications, including robotic manipulation (Huang et al., 2022; Shridhar et al., 2023) and embodied navigation (Qiao et al., 2025; Huang et al., 2023). To address this, recent works such as SpatialVLM (Chen et al., 2024a) and SpatialRGPT (Cheng et al., 2024) typically propose scalable data synthesis and curation pipelines to strengthen *single-view* spatial reasoning capabilities through large-scale pre-training. Despite these advances, more recently, Yin et al. (2025); Huang et al. (2024); Ma et al. (2023) demonstrates that current MLLMs and such approaches still struggle with geometric understanding and perspective-taking in *multi-view* settings. In this work, we are among the first to tackle this challenge, and we propose a novel zero-shot visual programming framework called pySpatial that systematically combines and applies various spatial tools, enabling models to explicitly reason in 3D and solve diverse spatial tasks.

**3D Reconstruction**. Classical 3D reconstruction methods, such as Structure-from-Motion (Schonberger & Frahm, 2016), typically involve multiple stages and often rely on time-consuming optimization pipelines. More recently, feed-forward 3D reconstruction approaches such as DUSt3R (Wang et al., 2024b), MASt3R (Leroy et al., 2024), CUT3R (Wang et al., 2025b) and VGGT (Wang et al., 2025a) leverage large-scale 3D pre-training and vision transformers to directly predict pixel-aligned 3D point maps. These data-driven methods demonstrate strong generalizability, even in scenarios without overlapping views. Building on this progress, subsequent works have extended feed-forward 3D reconstruction to applications in neural rendering (Charatan et al., 2024), SLAM (Maggio et al., 2025), and dynamic reconstruction (Lin et al., 2025; Luo et al., 2025).

**Modular Visual Reasoning**. To enhance compositional multi-modal understanding, recent advances treats vision specialists (such as GroundingDINO (Liu et al., 2024) and SAM (Ravi et al., 2025)) as symbolic operators and composes them to solve complex reasoning problems. Representative works such as Visual ChatGPT (Wu et al., 2023), MM-REACT (Yang et al., 2023), and HuggingGPT (Shen et al., 2023) follow this direction by integrating LLMs with predefined toolchains to process multi-modal inputs. Building on this idea, VisProg (Gupta & Kembhavi, 2023) and ViperGPT (Surís et al., 2023) introduce *visual programming* that extends this paradigm by prompting MLLMs to generate executable Python programs that call a set of visual parsers through predefined APIs. More recently, VADAR (Marsili et al., 2025) introduces the visual programming paradigm for single-view spatial reasoning tasks with an adaptive API design. In contrast, our pySpatial introduces a framework explicitly designed for multi-view spatial reasoning, equipping models with compositional 3D tools to handle diverse and complex spatial scenarios.

## 3 METHOD

In this section, we present pySpatial, a visual programming framework that enables MLLMs to reason explicitly in 3D space by generating and executing visual programs that orchestrate multiple spatial tools to address diverse spatial reasoning tasks. We also describe the framework design, including the pySpatial API signatures and the spatial tools it employs.

## 3.1 PROBLEM FORMULATION

We consider a setting where an MLLM $\mathcal{M}$ is provided with an image sequence $\mathcal{I} = \{I_n\}_{n=1}^N$, where each view has resolution $H \times W$ and captures partial observations of a 3D scene, along with a natural-language query $q$ concerning spatial relations between objects or camera movements. The objective is to produce the correct response $r^*$ from the answer space $\mathcal{A}$ that answers the query.

As introduced in Section 1, we convert the limited 2D views into an *explorable 3D scene* via feed-forward reconstruction. This yields consistent depth estimates $D$, camera intrinsics $K \in \mathbb{R}^{3 \times 3}$, and extrinsics $\mathbf{G} \in \mathrm{SE}(3)$ for each frame. Together, these quantities define a point cloud $\mathcal{P}$ in world coordinates, which serves as the geometric basis for downstream reasoning.

In addition, we adopt a program synthesis-perspective following Surís et al. (2023). Given an input of an image sequence and a query $(\mathcal{I}, q)$, a code agent $\mathcal{F}$ generates a Python program $z$ that invokes a set of spatial tools through a well-defined API. The program is executed by an interpreter $\mathcal{E}$ to produce an intermediate output $O$, which may take the form of text, a single image, or a list of images depending on the program $z$. This output provides direct visual evidence to support answering the query. For instance, when the query asks, "*what is behind me if I am at view 3*," the program renders a new view by rotating the camera 180° at the specified viewpoint. Finally, the MLLM $\mathcal{M}$ integrates both the original visual inputs and the program outputs to generate the final response $r \in \mathcal{A}$.

## 3.2 SPATIAL TOOLS AND API

To guide the MLLMs to explicitly reason in 3D space, we introduce various spatial tools such as 3D reconstruction, camera description, and novel view synthesis. We provide the `pySpatial` API signatures in Code 1 and the details of each tool are described in the following sections.

**3D Reconstruction**. We adopt two feed-forward reconstructions depending on the specific task requirement. For metric-scale scenes, we use CUT3R (Wang et al., 2025b), which returns depth in real-world units. When relative distance in normalized unit space suffices, we adopt VGGT (Wang et al., 2025a) for its generalizability.

Formally, each pixel $\mathbf{p}_i \in \mathbb{R}^2$ in a view $I_n$ with predicted depth $D_n(\mathbf{p}_i)$ is back-projected into the camera coordinate system using the intrinsics $K$, and then transformed into world coordinates via the estimated pose $\mathbf{G}_n \in \mathrm{SE}(3)$:

$$\mathbf{X}_i = \mathbf{G}_n^{-1} \pi^{-1} \big( \mathbf{p}_i, D_n(\mathbf{p}_i), K^{-1} \big), \quad (1)$$

where $\pi^{-1}$ denotes the back-projection from image coordinates to the 3D point in the camera frame. We get the point cloud $\mathcal{P}$ in the world space by concatenating $\mathbf{X}_i$ for all pixels in all frames.

**Camera Description**. We translate raw camera pose matrices into natural language labels to make egocentric motion interpretable

```python
class pySpatial:
    """pySptial interface for 3D vision tools."""

    def reconstruct(scene: Scene):
        # 3D reconstruction from scene images.

    def describe_camera_motion(recon:
      Reconstruction):
        # Describe camera motion from
      reconstruction results.

    def synthesize_novel_view(recon: Reconstruction
      , new_camera_pose):
        # Generate novel view synthesis from
      reconstruction results.

    def rotate_right(extrinsic, angle=45):
        # Rotate camera pose to the right, rotate
      45 degree by default

    def rotate_left(extrinsic, angle=45):
        # Rotate camera pose to the left rotate 45
      degree by default

    def move_forward(extrinsic, distance=0.3):
        # Move camera pose forward, a default
      distance is provided

    def move_backward(extrinsic, distance=0.3):
        # Move camera pose backward, a default
      distance is provided

    def turn_around(extrinsic):
        # Turn camera pose around 180 degrees
```

Code 1: `pySpatial` **API signatures**.

to the language model. Each pose is represented by an extrinsic matrix $\mathbf{G} = [\mathbf{R} \mid \mathbf{t}] \in \mathbb{R}^{3 \times 4}$, which maps world points into the camera frame. The corresponding camera center in world coordinates is $\mathbf{C} = -\mathbf{R}^\top \mathbf{t}$. Given two poses $(\mathbf{R}_1, \mathbf{t}_1)$ and $(\mathbf{R}_2, \mathbf{t}_2)$, the displacement in world coordinates is $\Delta \mathbf{C}_w = \mathbf{C}_2 - \mathbf{C}_1$. We then express this displacement in the first camera's frame as $\Delta \mathbf{C}_1 = \mathbf{R}_1 \Delta \mathbf{C}_w$. Restricting the displacement to the horizontal plane, we compute the yaw angle $\theta = \mathrm{atan2}(d_x, d_z) \cdot 180/\pi$, where $(d_x, d_z)$ are the $x$ and $z$ components of $\Delta \mathbf{C}_1$. The angle is discretized into eight canonical motion categories (forward, backward, left, right, and four diagonals), yielding a compact natural-language description of egocentric movement.

**Novel View Synthesis**. To facilitate active exploration of the reconstructed 3D scene, we enable the agent to render *novel views* from arbitrary camera poses. Given a point cloud $\mathcal{P}$ and a

world-to-camera transformation $\mathbf{G} = [\mathbf{R} \mid \mathbf{t}] \in \mathbb{R}^{3\times4}$, we rasterize $\mathcal{P}$ into an RGB image with respect to $\mathbf{G}$ and the corresponding camera intrinsics $\mathbf{K}$. The agent can then issue high-level actions such as `rotate_left` and `turn_around`, which are implemented as yaw rotations about the world $y$-axis by angle $\phi$. The updated camera pose $\mathbf{G}'$ is obtained by applying the rotation to the camera-to-world transform and inverting back to world-to-camera form. This operation provides interactive visual feedback that supports explicit spatial reasoning.

### 3.3 3D VISUAL PROGRAMMING

**Program Generation**. Given a query $q$, the code agent $\mathcal{F}$ synthesizes a Python program $z = \mathcal{F}(q)$ that composes function calls specified in the `pySpatial` API. By default, we use GPT-4o, a strong MLLM baseline that has demonstrated effectiveness in code generation, as it has been trained on Internet-scale Python code data. Note that the agent interacts only with the public interface (e.g., `reconstruct`, `rotate_right`, `synthesize_novel_view`) and has no access to internal implementation details such as model weights, file I/O, or rendering backends. This abstraction separates high-level reasoning from low-level execution. We also provide default parameters for public interface regarding rotation and movement, *i.e.* 45 for rotation, and 0.3 for movement, as specified in Code 1. We guide program synthesis using in-context examples, where the prompts include interface documentation and query–code pairs without ground-truth answers. In addition, we leverage structured outputs to first enable free-form natural language reasoning, followed by the synthesis of Python code. The generated Python code, or visual program, acts as an explicit intermediate representation that encodes a sequence of tool invocations. It is inherently interpretable, as researchers can readily inspect, debug, or modify the generated program, and composable, enabling seamless integration with additional tools or downstream reasoning modules. Once constructed, the program is executed by the interpreter to produce concrete spatial operations.

**Program Execution**. At execution time, the synthesized program $z$ is executed by a Python interpreter $\mathcal{E}$ over the input image sequence $\mathcal{I}$, yielding an intermediate output $O = \mathcal{E}(z, \mathcal{I})$. Depending on the query, the output $O$ may take the form of text, a single image, or a sequence of rendered views. This intermediate output provides an explicit grounding of the program's reasoning steps in observable evidence. In the final stage, a MLLM $\mathcal{M}$ integrates the original image sequence $\mathcal{I}$, the program output $O$, and the natural language query $q$ to generate the final response $r = \mathcal{M}(\mathcal{I}, O, q)$.

## 4 EXPERIMENTS

In this section, we assess the effectiveness of `pySpatial` on MINDCUBE (Yin et al., 2025) and OMNI3D-BENCH (Marsili et al., 2025), comparing it with existing state-of-the-art approaches.

### 4.1 EXPERIMENTAL SETTINGS

**Benchmarks**. We mainly evaluate our framework on the MINDCUBE (Yin et al., 2025), which is designed to probe the spatial reasoning capabilities of MLLMs under limited views. Specifically, MINDCUBE contains over 21,000 spatial question–answer pairs grounded in 3,268 multi-view indoor scenes, spanning three canonical camera motion types: rotation, around, and among. We also evaluate on MINDCUBE-1k, a subset of MINDCUBE with 1,050 questions, specifically designed for evaluation purposes. In addition, following prior work (Marsili et al., 2025), we also evaluate our framework on OMNI3D-BENCH, a single-view spatial reasoning benchmark, to examine whether our visual programming approach can generalize beyond multi-view settings.

**Baselines**. We compare the performance of `pySpatial` against four categories of existing baselines: (1) open-weight multi-image MLLMs, such as LLaVA-OneVision-7B (Li et al., 2025) and Qwen2.5-VL-3B-Instruct (Bai et al., 2025); (2) proprietary MLLMs, including GPT-4o, GPT-4.1-mini, and Claude-4-Sonnet; (3) specialized spatial models, such as Space-Qwen (Chen et al., 2024a) and VLM-3R (Fan et al., 2025), and (4) prior visual programming approaches such as ViperGPT (Surís et al., 2023), VisProg (Gupta & Kembhavi, 2023), and VADAR (Marsili et al., 2025).

**Implementation Details**. By default, we follow prior visual programming work (Marsili et al., 2025) to leverage GPT-4o as the code agent to generate Python programs and produce final responses to queries. We use VGGT (Wang et al., 2025a) as 3D reconstruction model on the MINDCUBE and OMNI3D-BENCH benchmarks. For real-world navigation, we use CUT3R (Wang et al., 2025b), which provides metric-scale reconstructions rather than normalized outputs. For point cloud rasteri-

Table 1: **Performance comparison on the full MINDCUBE (Yin et al., 2025) dataset**. The best results are shown in **bold**, and the second-best are underlined. Note that we implement pySpatial using GPT-4.1-mini as the code agent for this dataset due to budget constraints.

| Method | Reference | Overall | Rotation | Among | Around |
|---|---|---|---|---|---|
| *Baseline* | | | | | |
| Random (chance) | - | 32.35 | 36.36 | 32.29 | 30.66 |
| Random (frequency) | - | 33.02 | 38.30 | 32.66 | 35.79 |
| *Open-Weight Multi-Image Models* | | | | | |
| LLaVA-OneVision-7B | Li et al. (2025) | 47.43 | 36.45 | 48.42 | 44.09 |
| LLaVA-Video-Qwen-7B | Zhang et al. (2025c) | 41.96 | 35.71 | 43.55 | 30.12 |
| mPLUG-Owl3-7B-241101 | Ye et al. (2025) | 44.85 | 37.84 | 47.11 | 26.91 |
| InternVL2.5-8B | Chen et al. (2024b) | 18.68 | 36.45 | 18.20 | 13.11 |
| Qwen2.5-VL-7B-Instruct | Bai et al. (2025) | 29.26 | 38.76 | 29.50 | 21.35 |
| Qwen2.5-VL-3B-Instruct | Bai et al. (2025) | 33.21 | 37.37 | 33.26 | 30.34 |
| DeepSeek-VL2-Small | Lu et al. (2024) | 47.62 | 37.00 | 50.38 | 26.91 |
| *Proprietary Models* | | | | | |
| GPT-4o | OpenAI (2024) | 38.81 | 32.65 | 40.17 | 29.16 |
| GPT-4.1-mini | OpenAI (2025) | 45.62 | 37.84 | 47.22 | 34.56 |
| Claude-4-Sonnet | Anthropic (2025) | 44.75 | **48.42** | 44.21 | 47.62 |
| *Specialized Spatial Models* | | | | | |
| RoboBrain | Ji et al. (2025) | 37.38 | 35.80 | 38.28 | 29.53 |
| SpaceMantis | Chen et al. (2024a) | 22.81 | 37.65 | 21.26 | 29.32 |
| Spatial-MLLM | Wu et al. (2025) | 32.06 | 38.39 | 20.92 | 32.82 |
| Space-Qwen | Chen et al. (2024a) | 33.28 | 38.02 | 33.71 | 26.32 |
| VLM-3R | Fan et al. (2025) | 42.09 | 36.73 | 44.22 | 24.45 |
| pySpatial (**Ours**) | - | **58.56** | 43.20 | **60.54** | **48.10** |

Table 2: **Performance comparison on the MINDCUBE-1k (Yin et al., 2025) dataset**. The evaluated mental models (Yin et al., 2025) are based on Qwen2.5-VL-3B-Instruct (Bai et al., 2025). VADAR w/ Recon. denotes that we implement VADAR with our 3D reconstruction module. The best results are highlighted in **bold**, and the second-best are underlined.

| Method | Reference | Overall | Rotation | Among | Around |
|---|---|---|---|---|---|
| *Baseline Models* | | | | | |
| Qwen2.5-VL-3B-Instruct | Bai et al. (2025) | 37.81 | 34.00 | 36.00 | 45.20 |
| GPT-4o | OpenAI (2024) | 42.29 | 35.00 | 43.00 | 46.40 |
| *Spatial Mental Models* | | | | | |
| Chain-of-Thought | | 40.48 | 32.00 | 36.00 | 58.00 |
| View Interpolation | Yin et al. (2025) | 37.81 | 35.50 | 36.50 | 42.80 |
| Cognitive Map | | 41.43 | 37.00 | 41.67 | 44.40 |
| *Visual Programming Approaches* | | | | | |
| ViperGPT | Surís et al. (2023) | 36.95 | 20.50 | 41.00 | 40.40 |
| VADAR | Marsili et al. (2025) | 40.76 | 33.50 | 40.67 | 46.80 |
| VADAR w/ Recon. | - | 35.62 | 31.00 | 36.83 | 36.40 |
| pySpatial (**Ours**) | - | **62.35** ± 1.18 | **41.83** ± 2.34 | **64.89** ± 2.60 | **72.67** ± 3.30 |

zation, we use Open3D (Zhou et al., 2018) to render novel views. All experiments are conducted on a single NVIDIA A6000 Ada GPU. We provide full implementation details of pySpatial, along with the prompts used, in Appendix B and C. Code will be made publicly available upon acceptance.

## 4.2 QUANTITATIVE RESULTS

**Results on MINDCUBE**. We first perform comprehensive evaluations of pySpatial on the challenging MINDCUBE benchmark to rigorously assess its effectiveness in multi-view spatial reasoning. Table 1 summarizes the results in comparison with baseline approaches. Overall, pySpatial achieves a clear performance margin over all categories of baselines. Specifically, it reaches an overall accuracy of 58.56%, outperforming the best open-weight model DeepSeek-VL2-Small by 10.94%, and surpassing the strongest proprietary model GPT-4.1-mini by 12.94%. On the *Among* category, which requires reasoning over how the central object relates to all surrounding objects, pySpatial achieves 60.54%, substantially outperforming all baselines, none of which exceed 50%. Remarkably, pySpatial also outperforms VLM-3R (Fan et al., 2025), which leverages

Table 3: **Performance comparison on OMNI3D-BENCH**. Following VADAR (Marsili et al., 2025), We report mean relative accuracy (Yang et al., 2025) for the *numeric (other)* and accuracy for the other category. The best results are shown in **bold**, and the second-best are underlined.

| Method | Reference | numeric (ct) | numeric (other) | y/n | multi-choice | Total |
|---|---|---|---|---|---|---|
| *Baseline Models* | | | | | | |
| GPT-4o | OpenAI (2024) | **28.1** | 35.5 | 66.7 | 57.2 | 42.9 |
| Claude3.5-Sonnet | Anthropic (2024) | 22.4 | 20.6 | 62.2 | 50.6 | 32.2 |
| Llama-3.2 | Meta (2024) | 24.3 | 19.3 | 47.5 | 27.4 | 25.6 |
| Gemini1.5-Pro | Google (2024) | 25.2 | 28.1 | 46.2 | 37.6 | 32.0 |
| SpaceMantis | Chen et al. (2024a) | 20.0 | 21.7 | 50.6 | 48.2 | 30.3 |
| *Visual Programming Approaches* | | | | | | |
| ViperGPT | Surís et al. (2023) | 20.0 | 15.4 | 56.0 | 42.4 | 26.7 |
| VisProg | Gupta & Kembhavi (2023) | 2.9 | 0.9 | 54.7 | 25.9 | 13.5 |
| VADAR | Marsili et al. (2025) | 21.7 | 35.5 | 56.0 | **57.6** | 40.4 |
| pySpatial (**Ours**) | - | 22.9 | **38.6** | **72.0** | 54.7 | **44.2** |

CUT3R (Wang et al., 2025b) as the 3D encoder and is fine-tuned on synthetic spatial reasoning data, by 16.5%, despite operating entirely in a zero-shot setting. These results demonstrate that pySpatial generalizes well across diverse task categories on MINDCUBE. By explicitly decomposing spatial reasoning into modular tool calls, our approach provides a stronger inductive bias than both open-weight and proprietary MLLMs, including those specialized for spatial reasoning.

**Results on MINDCUBE-1k**. Table 2 compares pySpatial against approaches based on implicit mental modeling (Yin et al., 2025) (*e.g.*, chain-of-thought reasoning, cognitive maps) and prior visual programming agents (*e.g.*, ViperGPT, VADAR) on MINDCUBE-1k. We have the following key observations: (1) Spatial mental models (Yin et al., 2025), which rely on the implicit imagination mechanisms of MLLMs for spatial reasoning, yield only limited performance gains, whereas pySpatial outperforms each of them by roughly 20%; (2) Our pySpatial substantially outperforms existing visual programming approaches, achieving, for example, a 21.9% improvement over VADAR. Notably, pySpatial also surpasses VADAR w/ Recon., where we re-implement VADAR using our 3D reconstruction module. This result demonstrates that even when equipped with 3D information, VADAR's adaptive API design remains unreliable and lacks robustness for reasoning in 3D space. These results validate the superior effectiveness of pySpatial over existing baselines, demonstrating the advantages of enabling explicit 3D reasoning for multi-view spatial reasoning.

**Results on OMNI3D-BENCH**. We further evaluate pySpatial on the recent single-view spatial reasoning benchmark OMNI3D-BENCH, demonstrating that our framework generalizes effectively to single-view settings and provides consistent improvements across task categories. Table 3 shows results on OMNI3D-BENCH, where we follow the evaluation protocol of VADAR (Marsili et al., 2025): mean relative accuracy (MRA) is reported for the *numeric (other)* subtask, and standard accuracy is used for the remaining categories. Our pySpatial outperforms prior visual programming approaches, achieving gains of 3.8% over VADAR and 17.5% over ViperGPT, and sets a new overall state-of-the-art on OMNI3D-BENCH. Notably, pySpatial also surpasses GPT-4o on the total score, underscoring that our visual programming framework provides benefits even over advanced proprietary MLLMs. This result highlights the broad generalizability of pySpatial: even in single-view settings where geometric cues are less apparent, explicitly invoking 3D functions through the code agent continues to enhance spatial reasoning.

### 4.3 QUALITATIVE RESULTS

To further illustrate the capabilities of our pySpatial framework, we conduct qualitative experiments on representative examples from the MINDCUBE benchmark. As shown in Figure 2, each query is paired with the generated 3D visual program, the reconstructed 3D scene, the program outputs, and the final response produced by pySpatial. These examples highlight how pySpatial enables MLLMs to reason explicitly within an explorable 3D scene reconstructed from sparse 2D inputs. By synthesizing executable and interpretable visual programs that perform operations such as camera translation, rotation, and novel view synthesis, the framework provides interpretable outputs that ground the reasoning process in geometric evidence. Across diverse spatial reasoning tasks, pySpatial produces responses that closely align with ground-truth annotations, highlighting the effectiveness of our approach. It is worth noting that the generated 3D visual programs

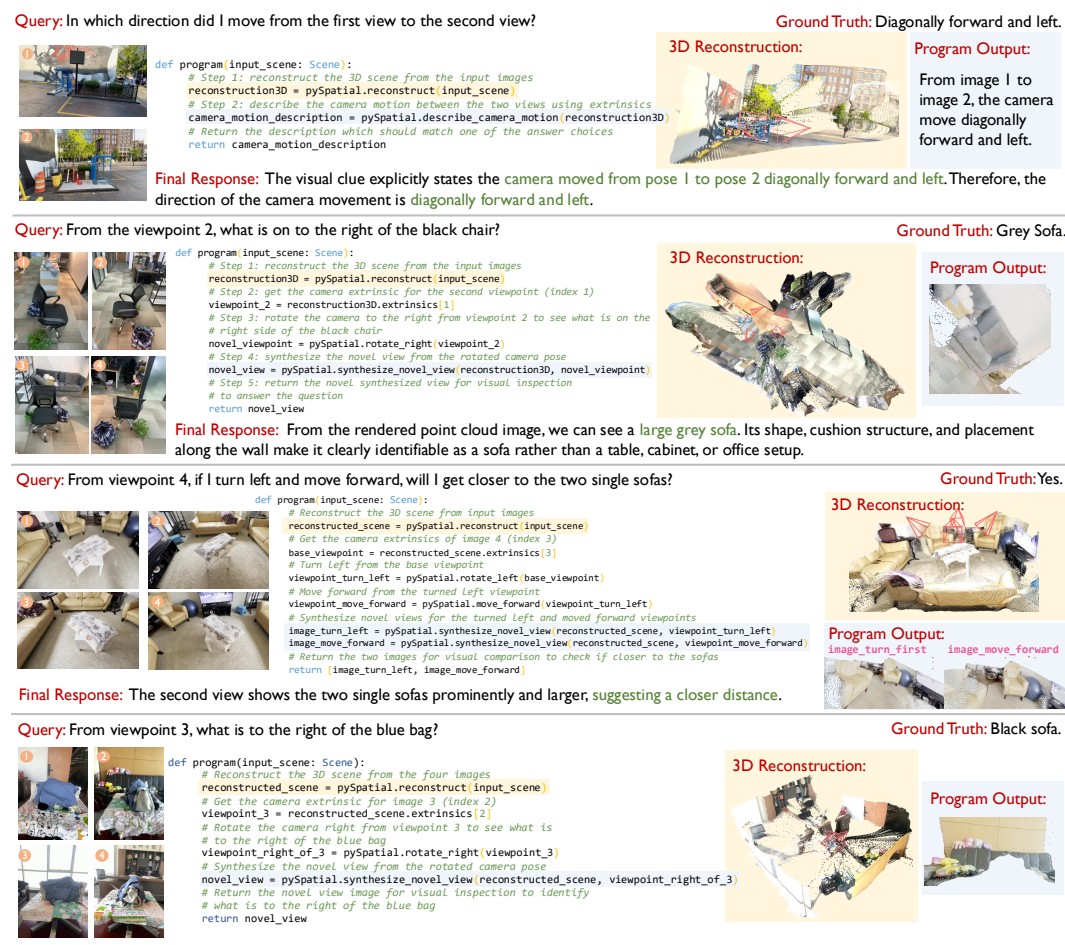

Figure 2: **Qualitative results on four representative examples from MINDCUBE.** We show that pySpatial enables MLLMs to explicitly reason within the reconstructed explorable 3D scene, effectively addressing diverse spatial reasoning tasks through interpretable and executable 3D visual programs. Figure A1 further illustrates that pySpatial is capable of composing executable 3D visual programs with control flow constructs (*e.g.*, *for*-loops), allowing it to robustly address a wide range of spatial reasoning tasks. Best viewed when zoomed in.

include well-structured comments that capture the reasoning process of pySpatial, thereby providing transparency and interpretability that researchers can readily verify, debug, or modify.

## 4.4 REAL-WORLD ROBOT NAVIGATION

To test the potential of real-world deployment using purely MLLMs, we employ a quadrupedal robot with a velocity-tracking controller in a 50 m² two-room laboratory. In this setup, the MLLM generates high-level position commands, which are manually converted into temporal velocity targets that the controller tracks, enabling the robot to navigate from an initial pose to a target object (a mushroom toy). From limited 2D views, pySpatial reconstructs an explorable 3D scene, infers camera poses via visual programming, and generates a structured motion plan for the robot to execute.

As shown in Figure 3, our pySpatial successfully guides the robot through doorways, make correct turns, and finally toward the correct goal location. Notably, the MLLM baseline GPT-4.1 struggles to resolve relative direction such as left–right and fails to provide absolute metric distance estimates, leading to navigation errors. In contrast, our agent outputs precise rotations and translations that align with real-world execution, resulting in reliable task completion. This experiment demonstrates that our approach not only produces coherent spatial reasoning in question answering benchmarks, but also transfers effectively to physical robotic platforms for complex indoor navigation tasks.

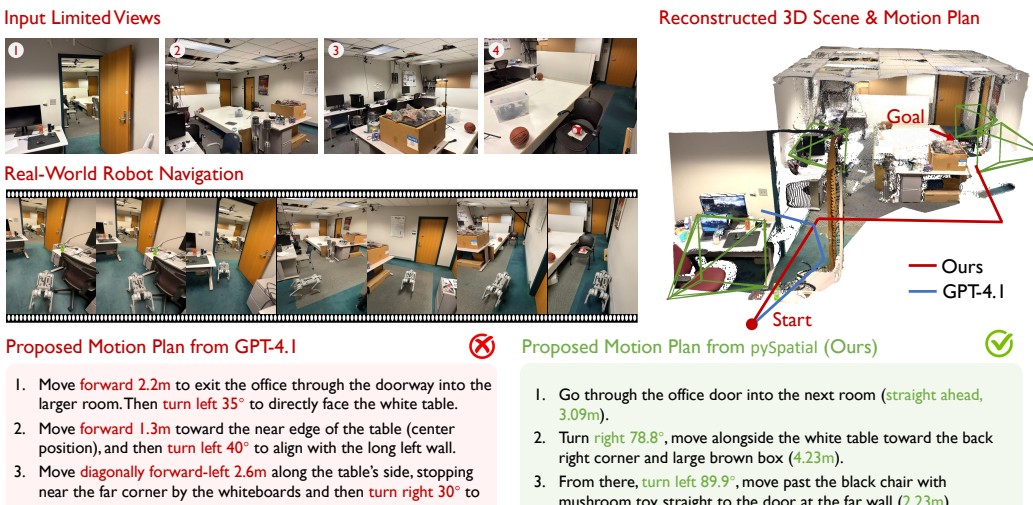

**Query:** Given a quadrupedal robot initialized at the pose corresponding to image 1, generate a task specification and motion plan for navigating to the mushroom toy at Image 4.

**Input Limited Views**

**Reconstructed 3D Scene & Motion Plan**

**Real-World Robot Navigation**

**Proposed Motion Plan from GPT-4.1** ⊗

1. Move forward 2.2m to exit the office through the doorway into the larger room. Then turn left 35° to directly face the white table.
2. Move forward 1.3m toward the near edge of the table (center position), and then turn left 40° to align with the long left wall.
3. Move diagonally forward-left 2.6m along the table's side, stopping near the far corner by the whiteboards and then turn right 30° to face the mushroom toy on the chair.

**Proposed Motion Plan from pySpatial (Ours)** ✓

1. Go through the office door into the next room (straight ahead, 3.09m).
2. Turn right 78.8°, move alongside the white table toward the back right corner and large brown box (4.23m).
3. From there, turn left 89.9°, move past the black chair with mushroom toy, straight to the door at the far wall (2.23m).

Figure 3: **Qualitative results on real-world robot navigation.** We deploy `pySpatial` on a Unitree-Go1 robot to navigate toward a target object (mushroom toy) using limited views as input. Compared to the GPT-4.1 baseline, which fails due to an incorrect initial turn and produces a collision-prone trajectory, `pySpatial` generates a geometrically consistent plan that successfully reaches the goal.

### 4.5 DISCUSSIONS

**Ablation Study on the Code Agent**. To ablate the effect of our code agent, we conduct experiments on the MINDCUBE-1k benchmark by comparing the performance of various MLLM baselines with and without integration of `pySpatial`. As summarized in Table 4, augmenting models with `pySpatial` consistently leads to substantial improvements across all tested MLLMs, including GPT-4o, GPT-4.1-mini, and GPT-4.1. For instance, GPT-4o improves from 42.3% to 62.7% overall accuracy, indicating that `pySpatial` generalizes across different MLLMs and effectively enhance spatial reasoning.

Table 4: **Ablation study on the code agent**. We report the accuracy on the MINDCUBE-1k dataset.

| Method | Overall | Rotation | Among | Around |
|---|---|---|---|---|
| GPT-4o | 42.29 | 35.00 | 43.00 | 46.40 |
| + pySpatial | 62.67 | 41.00 | 65.00 | 66.33 |
| GPT-4.1-mini | 43.34 | 36.00 | 45.00 | 44.80 |
| + pySpatial | 58.19 | 37.50 | 62.00 | 65.60 |
| GPT-4.1 | 44.67 | 35.50 | 45.33 | 50.40 |
| + pySpatial | 62.35 | 41.83 | 64.89 | 72.67 |

**Failure Case Analysis**. From the MINDCUBE benchmark, we select a representative subset of about 100 samples and conduct a manual analysis to identify the underlying sources of error in cases where the final response is incorrect. As shown in Figure 4, among the 39% of failure cases, only 6% are attributable to incorrectly generated visual programs that fail to address the query, validating the effectiveness of our overall programming pipeline. Beyond this, 20% of errors arise from the MLLMs at the final reasoning step, while 13% stem from limitations in the 3D reconstruction models, where the generated visual programs are correct but the program outputs do not provide useful information. These results also suggest that advances in 3D reconstruction and code generation models hold the potential to further enhance our performance.

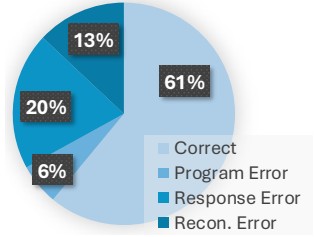

Figure 4: **Failure case study.** We manually examine the error sources in about 100 samples from MINDCUBE.

**Remarks on Efficiency**. Our `pySpatial` completes the MINDCUBE-1k benchmark in 2.17 hours on a single GPU using a single thread for 1,050 queries, averaging 7.45 seconds per query. As the breakdown, code generation requires 2.41 seconds, program execution 2.14 seconds, and answer generation 2.90 seconds. For comparison, VADAR (Marsili et al., 2025) requires 17.25 seconds per query on average. These results demonstrate that our visual programming framework enhances the spatial reasoning capabilities of MLLMs while remaining efficient to deploy without excessive cost.

## 5 CONCLUSION

In this work, we present `pySpatial`, a visual programming framework that enhance spatial reasoning capabilities of MLLMs through zero-shot Python code generation. By composing functions such as 3D reconstruction and novel-view synthesis, `pySpatial` converts 2D image sequences into explorable 3D scenes, enabling explicit reasoning in 3D space. Experiments on the MINDCUBE and OMNI3D-BENCH benchmarks demonstrate that `pySpatial` consistently outperforms strong MLLM baselines, with gains of up to 12.94% on MINDCUBE compared to GPT-4.1-mini. Beyond benchmarks, real-world indoor navigation experiments further validate its practicality, showing that robots can successfully traverse complex environments using route plans generated by `pySpatial`.

## ACKNOWLEDGEMENT

This work has been funded in part by the Army Research Laboratory (ARL) award W911NF-23-2-0007 and W911QX-24-F-0049, DARPA award FA8750-23-2-1015, and ONR award N00014-23-1-2840. The author would like to thank Dr. Xiang Yue for valuable discussions during the project.

## REPRODUCIBILITY STATEMENT

We are committed to ensuring the reproducibility of our results. All the spatial tools used in this work are open-sourced, and the benchmark datasets we evaluate on are publicly available. We have provided detailed descriptions of our experimental setup and implementation details in Section 4 and Appendix to facilitate reproducibility. Code is publicly available at https://github.com/Zhanpeng1202/pySpatial.

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

# Explicit 3D Spatial Reasoning via Program Generation

## Appendix

In the appendix, we provide additional experimental results and implementation details of our `pySpatial` framework. The appendix is organized as follows:

- Section A provides more experimental results.

- Section B shows the API specification for `pySpatial`.

- Section C presents the prompt implementation for our agent.

- Section D disclosures the use of large language models.

## A  MORE EXPERIMENTAL RESULTS

### A.1  RESULTS ON MMSI-BENCH

We further evaluate our approach on MMSI-Bench and report the results in Table A1. We observe that pySpatial improves the overall MMSI-Bench performance by 6.4% on average, further demonstrating the effectiveness of our approach.

Table A1: Evaluation results on MMSI-Bench. Our `pySpatial` is based on GPT-4o.

| Models | Positional Relationship | | | | | | Attribute | | Motion | | MSR | Avg. |
|---|---|---|---|---|---|---|---|---|---|---|---|---|
| | Cam.–Cam. | Obj.–Obj. | Reg.–Reg. | Cam.–Obj. | Obj.–Reg. | Cam.–Reg. | Meas. | Appr. | Cam. | Obj. | – | |
| *Proprietary* | | | | | | | | | | | | |
| GPT-5 | 43.0 | 35.1 | 32.1 | 48.8 | 42.4 | 51.8 | 60.9 | **36.4** | 32.4 | 36.8 | **42.0** | **41.9** |
| o3 | **45.2** | **39.4** | 37.0 | 44.2 | 47.1 | **62.6** | 54.7 | 28.8 | 31.1 | 32.9 | 34.9 | 41.0 |
| GPT-4.5 | 34.4 | 29.8 | **39.5** | **51.2** | 47.1 | 55.4 | 39.1 | 33.3 | **41.9** | **40.8** | 36.4 | 40.3 |
| GPT-4.1 | 36.6 | 26.6 | 27.2 | 29.1 | 36.5 | 27.7 | 37.5 | 24.2 | 36.5 | 32.9 | 28.8 | 30.9 |
| GPT-4o | 34.4 | 24.5 | 23.5 | 19.8 | 37.6 | 27.7 | 32.8 | 31.8 | 35.1 | 36.8 | 30.8 | 30.3 |
| Gemini-2.5-Pro | 39.7 | 31.9 | **39.5** | 45.3 | 35.2 | 43.3 | 51.5 | 21.2 | 36.4 | 30.2 | 34.3 | 36.9 |
| *Open-source* | | | | | | | | | | | | |
| InternVL3-78B | 34.4 | 23.4 | 32.1 | 12.8 | 37.6 | 26.5 | 37.5 | 19.7 | **28.4** | 31.6 | 29.3 | 28.5 |
| InternVL2.5-78B | 23.7 | 22.3 | **39.5** | 29.1 | 31.8 | **42.2** | 35.9 | 19.7 | 17.6 | 26.3 | 27.3 | 28.5 |
| Qwen2.5-VL-72B | 25.8 | **34.0** | 34.6 | 23.3 | 34.1 | 36.1 | **45.3** | 27.3 | 27.0 | 30.3 | 27.3 | **30.7** |
| LLaVA-OneVision-72B | **43.0** | 31.9 | 33.3 | 30.2 | 37.6 | 38.6 | 28.1 | 19.7 | 13.5 | 32.9 | 15.7 | 28.4 |
| *Baseline* | | | | | | | | | | | | |
| GPT-4o | 34.4 | 24.5 | 23.5 | 19.8 | 37.6 | 27.7 | 32.8 | 31.8 | 35.1 | 36.8 | 30.8 | 30.3 |
| + `pySpatial` (**Ours**) | 51.6 | 28.7 | 27.2 | 20.9 | 41.2 | 38.6 | 46.9 | 39.4 | 46.0 | 38.2 | 36.4 | 37.3 |

### A.2  MORE ABLATION STUDIES

**Code Agents**. In Table A2, we compare the performance of our `pySpatial` framework when paired with different code agents. Across all categories, `pySpatial` consistently improves upon the base GPT-4o model, regardless of the underlying LLM used for code generation. Among the evaluated agents, GPT-4o achieves the strongest overall performance, reaching 62.67% accuracy, while Qwen3-Coder and DeepSeek-v3 deliver comparable results at 62.10% and 61.05%, respectively. Notably, Qwen3-Coder performs best on the *Around* category, whereas GPT-4o provides the most balanced improvements across all task types. These findings indicate that our framework is robust to the choice of code agent and that most of the performance gains stem from the 3D visual programming paradigm rather than the specific code LLM used.

**3D Reconstruction Backbones**. In Table A3, we compare the impact of different 3D reconstruction backbones on the performance of our `pySpatial` framework. All three backbones, including VGGT, Pi3, and CUT3R, lead to substantial improvements over the GPT-4o baseline, indicating that our 3D visual programming paradigm is robust to the choice of reconstruction method. Among them, Pi3 achieves the best overall performance (63.33%), with notable gains in the *Rotation* and *Among* categories. VGGT provides similarly strong results, while CUT3R performs slightly lower but still significantly surpasses the base model. These findings suggest that `pySpatial` can effectively leverage a range of modern reconstruction backbones, and its spatial reasoning improvements are not tied to a specific reconstruction architecture.

Table A2: Performance comparison of our `pySpatial` framework with different code agents.

| Method | Overall | Rotation | Among | Around |
|---|---|---|---|---|
| GPT-4o | 42.29 | 35.00 | 43.00 | 46.20 |
| pySpatial w/ GPT-4o | 62.67 | 41.00 | 66.33 | 71.20 |
| pySpatial w/ Qwen3-Coder | 62.10 | 40.00 | 64.50 | 74.00 |
| pySpatial w/ DeepSeek-v3 | 61.05 | 40.50 | 65.33 | 64.80 |

Table A3: Performance comparison of `pySpatial` using different 3D reconstruction backbones.

| Method | Overall | Rotation | Among | Around |
|---|---|---|---|---|
| GPT-4o | 42.29 | 35.00 | 43.00 | 46.20 |
| pySpatial w/ VGGT | 62.67 | 41.00 | 66.33 | 71.20 |
| pySpatial w/ Pi3 | 63.33 | 43.50 | 65.66 | 72.33 |
| pySpatial w/ CUT3R | 61.05 | 40.50 | 64.33 | 69.60 |

Table A4: Comparison of `pySpatial` with different numbers of in-context learning examples.

| Method | Overall | Rotation | Among | Around |
|---|---|---|---|---|
| GPT-4o | 42.29 | 35.00 | 43.00 | 46.20 |
| pySpatial w/ 0 examples | 53.62 | 32.50 | 55.16 | 66.80 |
| pySpatial w/ 2 examples | 62.67 | 41.00 | 66.33 | 71.20 |
| pySpatial w/ 4 examples | 63.14 | 46.00 | 64.50 | 73.60 |

**In-Context Learning Examples for Code Agents**. In Table A4, we examine how the number of in-context learning examples influences the performance of our `pySpatial` framework. We observe a clear upward trend: providing even a small number of examples substantially improves performance across all categories. Using 0 examples already offers a strong boost over the base GPT-4o model (53.62% vs. 42.29%), demonstrating that `pySpatial` can operate effectively even without demonstration guidance. Adding 2 examples leads to a significant further gain, reaching 62.67% overall accuracy. Increasing to 4 examples yields the best performance (63.14%), with notable improvements particularly in the *Rotation* and *Around* categories. These results suggest that `pySpatial` benefits from additional examples, but even minimal in-context supervision is sufficient to unlock strong spatial reasoning capabilities.

### A.3 ADDITIONAL QUALITATIVE RESULTS

We present more qualitative results from MINDCUBE in Figure A1. Beyond the linear program flow demonstrated in Figure 2, `pySpatial` is also capable of executing more expressive control-flow operations, including for-loops, conditionals, and lambda-style functional compositions, allowing it to construct complex multi-step 3D operations. These richer program structures enable `pySpatial` to iteratively explore candidate viewpoints, selectively apply geometric predicates, and adapt its reasoning based on scene-dependent constraints. Across the presented examples, the 3D visual programs generated by our code agent consistently guide `pySpatial` toward responses that closely align with the ground-truth spatial relationships, further validating the correctness, interpretability, and robustness of our 3D reasoning framework.

### A.4 ADDITIONAL REAL-WORLD EXPERIMENTS

We also include an additional qualitative example in a challenging dynamic scene, where the views are captured at different times and a person is moving through the environment, as shown in Figure A2. Despite the temporal inconsistency and the presence of dynamic elements, `pySpatial` remains effective: the reconstruction module robustly integrates the multi-view observations and preserves the stable structural cues needed for accurate spatial reasoning. In contrast, GPT-5 continues to struggle in this scenario, failing to generate a safe or correct navigation trajectory and often producing instructions that are incompatible with the underlying scene geometry. These extended real-world experiments further validate the robustness and practical effectiveness of our approach.

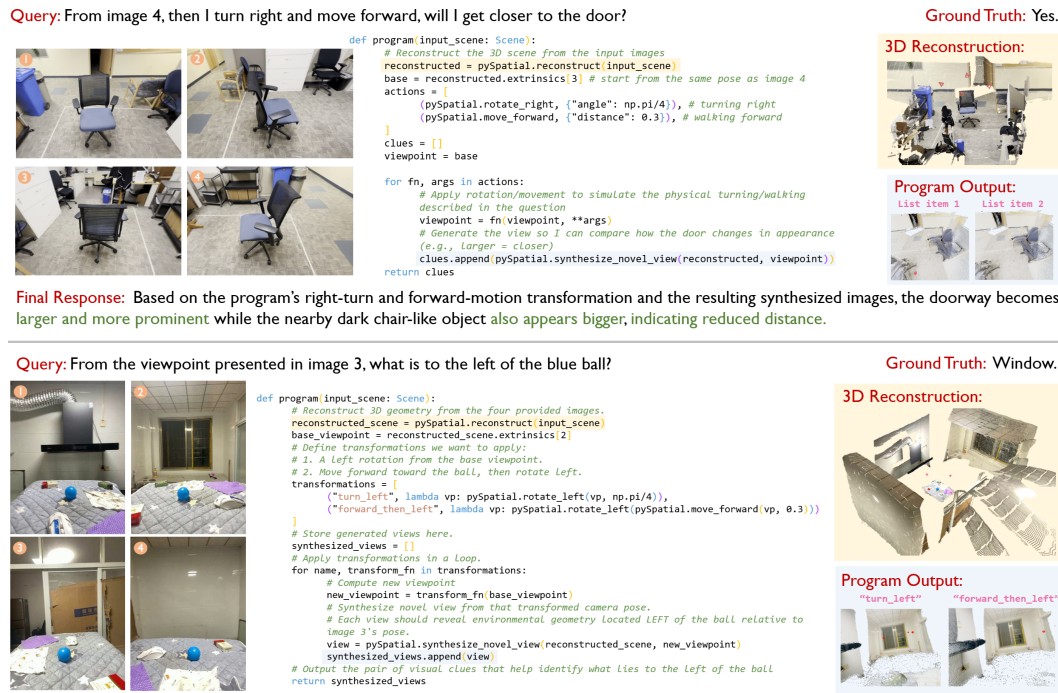

Figure A1: **More qualitative examples from MINDCUBE.** We show that `pySpatial` enables MLLMs to explicitly reason within a reconstructed, explorable 3D scene, allowing the model not only to interpret spatial structure but also to compose executable 3D visual programs with control flow, such as for-loops to robustly solve diverse spatial reasoning tasks.

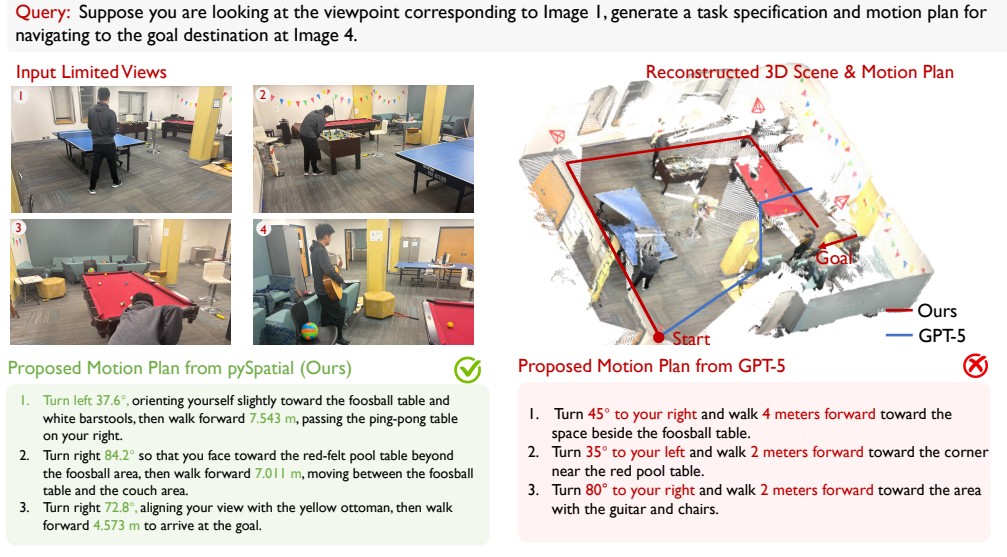

Figure A2: An additional qualitative real-world example in a challenging dynamic scene.

## B  IMPLEMENTATION FOR PYSPATIAL

A complete description of the API interface is provided in Code 2.

```python
import os
import glob
from typing import List, Union
from pathlib import Path

from tool.recontruct import reconstruct_3d
from tool.segment import segment_image, segment_automatic
from tool.estimate_depth import estimate_depth
from tool.camera_understanding import analyze_camera_trajectory
from tool.novel_view_synthesis import novel_view_synthesis, rotate_right, rotate_left,
    move_forward, move_backward, turn_around
import re

class Reconstruction:
    def __init__(self, point_cloud, extrinsics, intrinsics, colors=None):
        self.point_cloud = point_cloud
        self.extrinsics = extrinsics # list of 4 *4 numpy array
        self.intrinsics = intrinsics
        self.colors = colors  # Add colors attribute

class Scene:
    """Simple scene class that holds image data."""

    def __init__(self, path_to_images: Union[str, List[str]], question: str = "",
     scene_id: str = None):
        self.question = question
        self.scene_id = scene_id
        self.original_path = path_to_images  # Store original path for reconstruction
        self.images = self._load_images(path_to_images)
        self.reconstruction : Reconstruction = None
        self.code : str = None
        self.visual_clue = None

    def _load_images(self, path_to_images: Union[str, List[str]]) -> List[str]:
        """Load image paths from directory or list."""
        if isinstance(path_to_images, str):
            if os.path.isdir(path_to_images):
                # Check if this is a demo directory (contains .glb files)
                demo_path = Path(path_to_images)
                glb_files = list(demo_path.glob("*.glb"))

                if glb_files:
                    # This is a demo directory, load images from color/ subdirectory
                    color_dir = demo_path / "color"
                    if color_dir.exists():
                        image_extensions = ['*.png', '*.jpg', '*.jpeg']
                        images = []
                        for ext in image_extensions:
                            images.extend(glob.glob(os.path.join(str(color_dir), ext)))
                        return sorted(images)
                    else:
                        print(f"Warning: Demo directory detected but no color/
    subdirectory found in {path_to_images}")
                        return []
                else:
                    # Regular directory, load all images from directory
                    image_extensions = ['*.png', '*.jpg', '*.jpeg']
                    images = []
                    for ext in image_extensions:
                        images.extend(glob.glob(os.path.join(path_to_images, ext)))
                    return sorted(images)
            else:
                # Single image file
                return [path_to_images]
        else:
            # List of image paths
            return list(path_to_images)

class pySpatial:
    """Simple interface for 3D vision tools."""
    # we disable other function for now

    @staticmethod
```

```python
    def reconstruct(scene: Scene):
        """3D reconstruction from scene images."""

        # Check if this is a demo directory (contains .glb files)
        if isinstance(scene.original_path, str) and os.path.isdir(scene.original_path):
            demo_path = Path(scene.original_path)
            glb_files = list(demo_path.glob("*.glb"))

            if glb_files:
                # This is a demo directory, pass the directory path for demo data loading
                result = reconstruct_3d(scene.original_path, scene_id=scene.scene_id)
            else:
                # Regular reconstruction with image paths
                result = reconstruct_3d(scene.images, scene_id=scene.scene_id)
        else:
            # Regular reconstruction with image paths
            result = reconstruct_3d(scene.images, scene_id=scene.scene_id)

        # Convert the raw result dictionary to a Reconstruction object
        point_cloud = result.get('points', None)
        cameras = result.get('cameras', None)
        colors = result.get('colors', None)  # Get colors from result

        # Convert point cloud to numpy if it's a tensor
        if point_cloud is not None:
            if hasattr(point_cloud, 'cpu'):  # PyTorch tensor
                point_cloud = point_cloud.cpu().numpy()
            elif hasattr(point_cloud, 'numpy'):  # Other tensor types
                point_cloud = point_cloud.numpy()

        # Convert colors to numpy if it's a tensor
        if colors is not None:
            if hasattr(colors, 'cpu'):  # PyTorch tensor
                colors = colors.cpu().numpy()
            elif hasattr(colors, 'numpy'):  # Other tensor types
                colors = colors.numpy()

        # Extract extrinsics and intrinsics from cameras if available
        extrinsics = None
        intrinsics = None

        if cameras is not None:
            # Assume cameras contains extrinsic matrices
            extrinsics = cameras.cpu().numpy() if hasattr(cameras, 'cpu') else cameras

        # Create and return Reconstruction object with colors
        reconstruction = Reconstruction(point_cloud, extrinsics, intrinsics, colors)

        # Store the raw result for debugging
        reconstruction._raw_result = result

        return reconstruction

    @staticmethod
    def describe_camera_motion(recon: Reconstruction):
        """Describe camera motion from reconstruction results.
        Args:
        """
        extrinsics = recon.extrinsics
        return analyze_camera_trajectory(extrinsics)

    @staticmethod
    def synthesize_novel_view(recon: Reconstruction, new_camera_pose, width=512, height
=512, out_path=None):
        """Generate novel view synthesis from reconstruction results.
        Args:
            recon: Reconstruction object with point_cloud, extrinsics, intrinsics
            new_camera_pose: 3x4 or 4x4 extrinsic matrix for the new viewpoint
            width: output image width (default: 512)
            height: output image height (default: 512)
            out_path: output image path (default: None, returns image object if not
 provided)
        Returns:
            str or image: path to the rendered image if out_path provided, otherwise
 image object
        """
        return novel_view_synthesis(recon, new_camera_pose, width, height, out_path)

    @staticmethod
    def rotate_right(extrinsic, angle=None):
```

```python
        """Rotate camera pose to the right"""
            return rotate_right(extrinsic, angle)

    @staticmethod
    def rotate_left(extrinsic, angle=None):
        """Rotate camera pose to the left"""
            return rotate_left(extrinsic, angle)

    @staticmethod
    def move_forward(extrinsic, distance=None):
        """Move camera pose forward"""
            return move_forward(extrinsic, distance)

    @staticmethod
    def move_backward(extrinsic, distance=None):
        """Move camera pose backward"""
            return move_backward(extrinsic, distance)

    @staticmethod
    def turn_around(extrinsic):
        """Turn camera pose around 180 degrees"""
        return turn_around(extrinsic)

class Agent:
    def __init__(self, api_key: str = None):
        self.api_key = api_key or os.getenv('OPENAI_API_KEY')

    def generate_code(self, scene: Scene):
        from agent.codeAgent.query import generate_code_from_query
        return generate_code_from_query(scene, self.api_key)

    def parse_LLM_response(self, scene: Scene, response: str):
        """
        Extracts the first python code block (```python ... ```) from text.
        Returns the code as a string, or "" if not found.
        """
        from agent.codeAgent.execute import parse_LLM_response
        code = parse_LLM_response(response)
        scene.code = code
        return code

    def execute(self, scene: Scene):
        """
        Execute a code string with a scene and return the visual clue result.
        """
        # try:
        #     from agent.codeAgent.execute import execute_code
        #     program = execute_code(scene.code)

        #     visual_clue = program(scene)
        #     return visual_clue
        # except Exception as e:
        #     import traceback
        #     error_details = f"Execution failed: {str(e)}\nTraceback: {traceback.
    format_exc()}"
        #     # Store the error for detailed reporting
        #     self.last_execution_error = error_details
        #     return f"there is an error during code generation, no visual clue provided.
     Error: {str(e)}"

        from agent.codeAgent.execute import execute_code
        program = execute_code(scene.code)

        visual_clue = program(scene)
        return visual_clue

    def answer(self, scene: Scene, visual_clue):
        # answer the question with visual clue
        from agent.anwer import answer

        # Set the visual clue in the scene
        scene.visual_clue = visual_clue

        # Call the answer function with API key
        return answer(scene, self.api_key)
```

Code 2: **Full** `pySpatial` **API specification**.

## C    Implementation Details of the Agent Prompt in pySpatial

We present the prompts used in our experiments in the box below.

---

**AGENT PROMPT IN PYSPATIAL**

```python
task_description = """
    You are now asked to solve a spatial reasoning related problem.
    The input are image(s) and a natural langugae question that
    specifically designed to test your spatial reasoning ability.
    It is not trivial to solve these tasks directly as a vision
    langugae model. However, You have access to the following Python API:
"""

api_specification = """
    In the PySpatial API, we explicitly introduce the 3D inductive bias.
    We provide a Scene class that contains the image(s) and a question.
    Further, we also provide a 3D reconstruction process that can be
    used to generate a 3D point cloud and camera parameters.

    class Reconstruction:
        def __init__(self, point_cloud, extrinsics, intrinsics):
            self.point_cloud = point_cloud
            self.extrinsics = extrinsics
            self.intrinsics = intrinsics

    class Scene:
        "Simple scene class that holds image data."
        def __init__(self, path_to_images: Union[str, List[str]],
                                          question: str = ""):
            self.question = question
            self.images = self._load_images(path_to_images)
            self.reconstruction : Reconstruction = None

        def _load_images(self, path_to_images: Union[str, List[str]])
                                                -> List[str]:
            "Load image paths from directory or list."
            if isinstance(path_to_images, str):
                if os.path.isdir(path_to_images):
                    # Load all images from directory
                    image_extensions = ['*.png', '*.jpg', '*.jpeg']
                    images = []
                    for ext in image_extensions:
                        images.extend(glob.glob(os.path.join(
                                        path_to_images, ext)))
                    return sorted(images)
                else:
                    # Single image file
                    return [path_to_images]
            else:
                # List of image paths
                return list(path_to_images)

    class pySpatial:
        "Simple interface for 3D vision tools."
        # we disable other function for now
```

---

```
        @staticmethod
        def reconstruct(scene: Scene):
            "3D reconstruction from scene images."

            return reconstruct_3d(scene.images)

        @staticmethod
        def describe_camera_motion(recon: Reconstruction):
            "Describe camera motion from reconstruction results.
            Args:
            "
            extrinsics = recon.extrinsics
            return describe_camera_motion(extrinsics)

        @staticmethod
        def synthesize_novel_view(recon: Reconstruction,
                                  new_camera_pose):
            "Generate novel view synthesis from reconstruction results.
            Args:
            "
            return novel_view_synthesis(recon)

        # methods to manipulate camera pose
        def rotate_right(extrinsic, angle=np.pi/2):

        def rotate_left(extrinsic, angle=np.pi/2):

        def move_forward(extrinsic, distance=0.1):

        def move_backward(extrinsic, distance=0.1):

        def turn_around(extrinsic):

        @staticmethod
        def estimate_depth(image):
            return estimate_depth(image)
"""

# in-context learning exmaples
example_problems = """
    Problem 1:
    Question: "Based on these two views showing the same scene:
    in which direction did I move from the first view to the
    second view?
    A. Diagonally forward and left
    B. Directly right
    C. Directly left
    D. Diagonally forward and right"

    How to solve this problem?
    Step 1: we can easily find the ansewr with camera extrinsics.
    Step 2: therefore, we should first reconstruct the scene,
    and then use the camera extrinsics to find the answer.
    Step 3: it is still not trivial to directly get the answer
    from extrinsic matrix.
    Step 4: we can use the pySpatial.describe_camera_motion
```

```
    to get the answer.
    Next, write python code within the pySpatial API provided,
    then an agent will automatically collect the code
    I wrote and execute it.

    ```python
    def program(input_scene: Scene):
        reconstruction3D = pySpatial.reconstruct(input_scene)
        camera_motion = pySpatial.describe_camera_motion(
                    reconstruction3D)
        return camera_motion
    ```

    Step 5: After I get the visual clue from execution,
    I can easily match the answer:

    Problem 2:
    Based on these four images (image 1, 2, 3, and 4)
    showing the pink bottle from different viewpoints (front, left, back,
    and right),with each camera aligned with room walls and partially
    capturing the surroundings: If I am standing at the same spot and
    facing the same direction as shown in image 1, then I turn right
    and move forward, will I get closer to the pink plush toy
    and headboard?

    since we do not have the way to compare distance in 3D space,
    we can render two images, and use these two images as visual clue.
    ```python

    def program(input_scene: Scene):

        reconstructed_scene = pySpatial.reconstruct(input_scene)
        base_viewpoint = reconstructed_scene.extrinsics[0]
        # the image 1 indicates the 0th index in the array

        viewpoint_turn_right = pySpatial.rotate_right
                            (base_viewpoint)
        viewpoint_move_forward = pySpatial.move_forward
                                (viewpoint_turn_right)

        image_right = pySpatial.synthesize_novel_view
                    (reconstructed_scene, viewpoint_turn_right)
        image_forward = pySpatial.synthesize_novel_view
                     (reconstructed_scene, viewpoint_move_forward)

        # we should compare these two images, check if the object
          exists and if the distance is closer.
        visual_clue = [image_right, image_forward]
        return visual_clue
    ```
"""

code_generation_prompt = f"""
    Now please utilize the PySpatial API and write a python function
    to solve the problem.
    Noted that you can first do reasoning and then write the code.
    But the code should be wrapped in the ```python ``` block.
    Write a compact code block
```

```
    Also, the function written should be named as program
    and the input parameter should be a Scene object.
    for example,
    ```python
    def program(input_scene: Scene):
        ...
        return ...
    ```
    try to add simple comments to the code to explain your logic.

    Make sure to first reasoning, why we write program like this,
    becuase we have a pySpatial API that allows us to explore the 3D
    space, please first do a reasoning like (I want to know what is
    to the right of something, therefore I just render a novel view
    from that).
"""

# Prompt template for ReAct: ReAct: Synergizing Reasoning and Acting
in Language Models https://arxiv.org/abs/2210.03629

answer_background = f"""
    We are now solving a spatial reasoing problem.
    It is not trivial to solve these tasks directly as a vision language
    model.
    However, We have access to the following PySpatial API:
    {api_specification}

    We generate a python code based on the PySpatial API to solve
    this problem.
"""

answer_prompt = """
    Based on the code and the visual clue from the execution, answer
    the question.
"""

# Prompt for the answer without visual clue
without_visual_clue_background = """
    Solve this spatial reasoning problem based on the question
    and the image input.

    First, analyze the question, extract useful information from
    the question description, then try to answer it based on the
    useful visual information.

    Give your best guess if you cannot find the best answer.
"""
```

## D   THE USE OF LARGE LANGUAGE MODELS (LLMS)

We employed LLMs solely as an auxiliary tool to polish the writing of this manuscript. They were used to improve grammar, clarity, and readability, but no LLMs were involved in ideation, data analysis, experiment design, or result interpretation.

