# OpenReview forum: "pySpatial: Generating 3D Visual Programs for Zero-Shot Spatial Reasoning"
_ICLR.cc/2026/Conference — ICLR 2026 Poster_

### Official Review · Reviewer_FeNP · 2025-10-31

**Soundness:** 2
**Presentation:** 3
**Contribution:** 2
**Rating:** 4
**Confidence:** 4

**Summary:**

The paper proposes an agentic prompting method to improve spatial understanding tasks with explicit 3D representations and achieves suprior performance than existing MLLMs.

**Strengths:**

- The paper presents a novel agentic method, incorporating state-of-the-art 3D foundation models for spatial understanding tasks.
- The proposed method achieves suprior performance comparing to other methods.
- Real world application demonstrates the effectiveness of the whole system.

**Weaknesses:**

- Since the paper requires the model to explicitly reconstruct the 3D scenes, render novel views, and navigates inside the reconstructed scene. Is it sensitive to the reconstruction methods, *i.e.* incorporating 3D foundational models other than VGGT?

- Is the proposed method sensitive to the rendering quality? In Figure 2 we see holes in the rendered images. Is VLMs sensitive to those kind of inputs? Is there anyway to compensate the visual gap? Would gaussian splatting be a better choice?

- Is the method sensitive to the code agent choice? Table 4 mainly compares the performance of the GPT family. What about other VLMs, will coding VLMs perform better than general purpose VLMs?

**Questions:**

The explicit 3D modeling method for spatial understanding may be limited to static environments. While in real-world robot navigation tasks, how much will changes in the environment affect the overall performance?

---

> ### Author Response · Authors · 2025-11-22
> **Response to Reviewer FeNP**
>
> Dear Reviewer FeNP,
>
> We greatly appreciate your valuable feedback on our paper. We address the raised concerns and questions below.
>
> ---
>
>
> **Comment (1)**: “*Since the paper requires the model to explicitly reconstruct the 3D scenes, render novel views, and navigates inside the reconstructed scene. Is it sensitive to the reconstruction methods, i.e. incorporating 3D foundational models other than VGGT?*”
>
> **Response (1)**: Thanks for your insightful comments. Following your comments, we conducted a performance comparison of our approach using different reconstruction backends, including VGGT, Pi3, and CUT3R, under the same multi-view input setting. The results are reported as follows.
>
> | Method | Overall | rotation | Among | around |
> |----|----|---|--|---|
> | GPT-4o  | 42.29   | 35.00    | 43.00 | 46.20  |
> | pySpatial w/ VGGT   | 62.67   | 41.00    | 66.33 | 71.20  |
> | pySpatial w/ Pi3    | 63.33   | 43.50    | 65.66 | 72.33  |
> | pySpatial w/ CUT3R  | 61.05   | 40.50    | 64.33 | 69.60  |
>
> From the table, we can see that pySpatial achieves stable performance across different reconstruction backends, confirming that our method is robust with backbone selection. We have incorporated these results and discussions into Appendix A.2 of the revised paper.
>
> ---
>
>
> **Comment (2)**: “*Is the proposed method sensitive to the rendering quality? In Figure 2 we see holes in the rendered images. Is VLMs sensitive to those kind of inputs? Is there anyway to compensate the visual gap? Would gaussian splatting be a better choice?*”
>
> **Response (2)**:  We would like to clarify that the large holes observed in Figure 2 arise because the provided input views do not cover those spatial regions, and the missing geometry reflects true gaps in the captured observations rather than a limitation of the rendering backend. Given this input sparsity, any renderer (including point-cloud or Gaussian-splat approaches) would necessarily leave these areas under-explored.
>
> Empirically, we find that modern MLLMs are reasonably robust to such rendering artifacts. Despite the sparse regions, state-of-the-art MLLMs can still extract the key geometric cues needed for spatial reasoning, and their performance remains stable. This is also supported by our failure case study in Figure 4, where we show that aside from reconstruction and program-related errors, most of the generated responses remain correct despite the rendering artifacts.
>
> ---
>
>
> **Comment (3)**: “*Is the method sensitive to the code agent choice? Will coding VLMs perform better than general purpose VLMs?*”
>
> **Response (3)**: In addition to the proprietary models reported in the main paper, we have now included experiments with fully open-source code generation models. Specifically, we evaluate our pySpatial framework with Qwen3-Coder and DeepSeek-V3 under the same program synthesis setting. The results are as follows.
>
>
> | Method  | Overall | Rotation | Among | Around |
> |--|--|--|--|--|
> | GPT-4o | 42.29 | 35.00 | 43.00 | 46.20  |
> | pySpatial w/ GPT4o      | 62.67 | 41.00 | 66.33 | 71.20  |
> | pySpatial w/ Qwen3-Coder| 62.10 | 40.00 | 64.50 | 74.00  |
> | pySpatial w/ DeepSeek-v3| 61.05 | 40.50 | 65.33 | 64.80  |
>
> We observe that these open-source coding VLMs provide performance gains comparable to GPT-4o, indicating that our pySpatial framework is generally robust to the choice of code agent. We have included these results and the corresponding analysis in Appendix A.2 of the revised paper.
>
> ---
>
> **Comment (4)**: “*The explicit 3D modeling method for spatial understanding may be limited to static environments. While in real-world robot navigation tasks, how much will changes in the environment affect the overall performance?*”
>
> **Response (4)**: We agree that explicit 3D modeling tends to perform best in static environments. Empirically, however, our real-world experiments show that moderate scene changes (*e.g.*, object shifts across different views) do not significantly affect performance, and the robot can still successfully navigate to the goal. This is largely because the reconstruction models used in our framework, such as CUT3R, are able to handle such variations and can preserve the essential geometric structure required for spatial reasoning.
>
> To show this, we present an additional real-world qualitative example in Appendix A.4 of the revised paper, where the views are captured at different times and a person is moving through the environment. In this setting, pySpatial remains effective since the reconstruction module is able to handle the dynamic elements, while GPT-5 continues to fail to generate a safe or correct navigation trajectory. This extended real-world experiment further validates the robustness and practical effectiveness of our approach.
>
> ---
>
> We sincerely appreciate your thoughtful comments, which have helped us improve the paper. If you have any further questions or concerns, please don’t hesitate to let us know.
>
> Best,
>
> Authors

---

> > ### Comment · Reviewer_FeNP · 2025-11-27
> >
> > I thank the authors for their thorough response. My concerns have been addressed.

---

> > > ### Author Response · Authors · 2025-11-27
> > > **Thank you for your thoughtful reassessment and updated review**
> > >
> > > Thank you very much for taking the time to read our rebuttal. We truly appreciate your careful consideration and are glad our response addressed your concerns. And thank you for increasing the rating, we sincerely appreciate it.

---

### Official Review · Reviewer_Xvqg · 2025-11-01

**Soundness:** 3
**Presentation:** 3
**Contribution:** 2
**Rating:** 6
**Confidence:** 3

**Summary:**

The paper introduces pySpatial, a visual programming framework designed to enhance the 3D spatial reasoning abilities of multimodal large language models (MLLMs). While existing MLLMs perform well on general perception and reasoning tasks, they struggle with understanding spatial relationships in three-dimensional environments. pySpatial addresses this limitation by allowing MLLMs to generate and execute Python programs that call spatial tools to transform 2D image inputs into explorable 3D scenes. This enables models to reason explicitly over geometric structures without additional fine-tuning, operating entirely in a zero-shot manner. The method builds on recent advances in feed-forward 3D reconstruction and visual programming, integrating them into a unified, interpretable API for spatial reasoning. Experiments on the MINDCUBE and OMNI3D-BENCH benchmarks show that pySpatial outperforms strong MLLM baselines like GPT-4.1-mini. The framework also demonstrates practical effectiveness in real-world indoor navigation tasks, where a robot can successfully generate and follow route plans derived from pySpatial’s reasoning outputs.

**Strengths:**

1. The paper clearly identifies a relevant gap between current MLLMs’ implicit, imagination-based spatial reasoning and the need for explicit geometric grounding. The motivation is reasonable and reflects an active research direction in improving spatial understanding for embodied and multi-view settings.

2. The proposed visual programming framework is well-structured and methodologically sound. Its modular API design offers a clear and interpretable mechanism for integrating 3D reasoning tools with MLLMs. This contributes to transparency and reproducibility, though the overall paradigm follows existing visual programming approaches.

3. The framework demonstrates consistent zero-shot performance gains on both multi-view (MINDCUBE) and single-view (OMNI3D-BENCH) benchmarks. These improvements, achieved without any fine-tuning, indicate that the method is generally effective as a plug-and-play enhancement for existing MLLMs.

4. The experimental evaluation is thorough, with comparisons spanning open-weight models, proprietary systems, and prior visual programming baselines. Additional ablation and failure analyses help clarify the contribution of each component and provide a more complete understanding of the system’s strengths and limitations.

5. The experimental evaluation is thorough, with comparisons spanning open-weight models, proprietary systems, and prior visual programming baselines. Additional ablation and failure analyses help clarify the contribution of each component and provide a more complete understanding of the system’s strengths and limitations.

**Weaknesses:**

1. While the integration of 3D spatial reasoning within a visual programming framework is well-executed, the core concept of using generated Python code as an intermediate reasoning layer is not entirely novel. Prior works such as VisProg, ViperGPT, and VADAR have explored similar paradigms for visual reasoning. The novelty here primarily lies in extending this paradigm to 3D tools rather than introducing a fundamentally new reasoning mechanism. Consequently, the conceptual contribution may be perceived as incremental in scope.

2. The real-world navigation experiment, though interesting, is relatively limited in scale. It consists of a single scenario under controlled conditions. Additional experiments across varied environments, different robot embodiments, or more complex navigation tasks would significantly strengthen the claims regarding real-world applicability and robustness.

**Questions:**

1. Could pySpatial be integrated with or further enhance the spatial reasoning capabilities of specialized spatial MLLMs (e.g., VLM-3R or Space-Qwen)? Specifically, would combining explicit program-based reasoning with models already trained on 3D-augmented data yield additional improvements?

---

> ### Author Response · Authors · 2025-11-22
> **Response to Reviewer Xvqg**
>
> Dear Reviewer Xvqg,
>
> We greatly appreciate your valuable feedback and your positive recommendation of our work. We address the raised concerns and questions below.
>
> ---
>
> **Comment (1)**: “*While the integration of 3D spatial reasoning within a visual programming framework is well-executed, the core concept of using generated Python code as an intermediate reasoning layer is not entirely novel.*”
>
> **Response (1)**: Thank you for the thoughtful comment. We acknowledge that visual programming itself is not entirely novel, as earlier works have explored using generated code as an intermediate reasoning layer in 2D settings. Our motivation, however, is to meaningfully extend this paradigm into the 3D spatial domain, where the reasoning requirements are fundamentally different. Beyond basic image-level operations such as depth estimation or object detection, 3D reasoning additionally requires estimating camera motion, representing a consistent 3D scene, and performing geometry-aware queries that are not possible in 2D visual programming settings.
>
> It is also important to note that, to the best of our knowledge, this is the first work to address multi-view spatial reasoning through a visual programming paradigm. Our approach offers a new perspective compared to prior methods based on fine-tuning or learned spatial mental models, and it may provide useful insights for future research in this field.
>
> With our 3D visual programming framework, the system can handle a wide range of spatial reasoning tasks in a zero-shot manner, and across all benchmarks, pySpatial consistently achieves substantial improvements over both standard MLLMs and prior visual-programming-based approaches. These results validate the practical effectiveness of our contribution.
>
> ---
>
> **Comment (2)**: “*The real-world navigation experiment, though interesting, is relatively limited in scale.*”
>
> **Response (2)**: Thank you for your thoughtful feedback. Following your comments, we extended our real-world experiments by incorporating 10 additional scenarios that cover more diverse spatial layouts and object configurations. These added evaluations allow us to better assess the robustness of our approach under varying real-world conditions, and the results, as shown below, further support the generalization capability of our 3D reasoning framework.
>
> | Method | GPT-4o | GPT-5 | pySpatial |
> |-|-|-|-|
> | Success Rate |1 / 10 | 2 / 10 | 8 / 10 |
>
> We observe that both GPT-4o and GPT-5 are only able to handle relatively simple navigation scenarios, typically those involving two views with minimal viewpoint change and no required turns. In more challenging tasks, such as the multi-view indoor navigation example shown in Figure 3, GPT-series models consistently fail, often colliding at the very first corner of the expected trajectory. In contrast, pySpatial successfully completes these tasks by leveraging accurate 3D reconstruction and geometry-aware reasoning. In the most difficult cases, such as long hallways with repeated structural patterns or complex outdoor scenes with large open spaces, pySpatial also struggles and may fail to reach the goal. We will update our project website to include the corresponding video demonstrations.
>
> We have also included an additional qualitative example in a challenging dynamic scene to Appendix A.4 of the revised paper, where the views are captured at different times and a person is moving through the environment. In this setting, pySpatial remains effective since the reconstruction module is able to handle the dynamic elements, while GPT-5 continues to fail to generate a safe or correct navigation trajectory. These extended real-world experiments further validate the robustness and practical effectiveness of our approach.
>
> We agree that additional experiments across different robot embodiments would further strengthen the claims. However, due to hardware constraints, we currently have access to only a single embodied platform for physical evaluations. We hope the experiments above have adequately addressed your concerns and please let us know if you have further questions or concerns.
>
> ---
>
> **Comment (3)**: “*Could pySpatial be integrated with or further enhance the spatial reasoning capabilities of specialized spatial MLLMs (e.g., VLM-3R or Space-Qwen)?*”
>
> **Response (3)**: Thanks for your insightful comments. Theoretically, our pySpatial framework can be integrated with these spatial MLLMs to further enhance their spatial reasoning capabilities, as it is designed to be generally plug-and-play and training-free. However, due to the tight timeline, this experiment is still in progress. We are actively adapting the code and will share the results in a follow-up comment in the next few days.
>
> ---
>
> If you have any further questions or concerns, please don’t hesitate to let us know.
>
> Best,
>
> Authors

---

> > ### Author Response · Authors · 2025-11-28
> > **Follow-up Response to Reviewer Xvqg**
> >
> > Dear Reviewer Xvqg,
> >
> > Thank you again for your insightful review of our paper. We have completed the experiments applying our pySpatial framework to specialized spatial MLLMs, and we would like to provide a follow-up to our responses to **Comment (3)**.
> >
> > ---
> >
> > **Comment (3)**: “*Could pySpatial be integrated with or further enhance the spatial reasoning capabilities of specialized spatial MLLMs (e.g., VLM-3R or Space-Qwen)?*”
> >
> > **Response (3)**: Thank you for your insightful comments. Yes, our pySpatial framework can be applied to specialized spatial MLLMs to further enhance their spatial reasoning capabilities. Below, we report the results on the MindCube-1k benchmark:
> >
> > | Method               | Overall | Rotation | Among | Around |
> > |----------------------|---------|----------|--------|--------|
> > | GPT-4o           | 42.29   | 35.00    | 43.00 | 46.20  |
> > | VLM-3R           | 45.90  | 35.50    |46.83 | 52.00   |
> > | pySpatial w/ VLM-3R| 61.80   | 44.50    | 65.17 | 67.60  |
> >
> > As shown in the table, VLM-3R already achieves slightly higher performance than GPT-4o. With our pySpatial, VLM-3R can further improve by +15.9% accuracy on MindCube-1k, demonstrating that pySpatial is generally plug-and-play and can be seamlessly integrated with specialized models. We will incorporate these discussions and the new results into the revised paper.
> >
> > ---
> >
> > We hope our responses have adequately addressed your concerns. If you have any further questions or concerns, please don’t hesitate to let us know.
> >
> > Best,
> >
> > Authors

---

### Official Review · Reviewer_Rgza · 2025-11-01

**Soundness:** 2
**Presentation:** 3
**Contribution:** 2
**Rating:** 4
**Confidence:** 3

**Summary:**

This paper proposes pySpatial, a zero-shot visual programming method to perform multi-view spatial reasoning techniques. The model calls a set of pre-existing APIs best on the reasoning of MLLM to output the final answer for spatial reasoning tasks.

**Strengths:**

In general the paper is written quite clearly. The method is well described, and Figure 1 and is quite clear in terms of describing the differences with most spatial mental models and pySpatial. The model is compared on several datasets (MindCube, Omni3D-Bench) to show the effectiveness.

The problem of spatial reasoning with MLLMs is also an important and relevant task in the community.

**Weaknesses:**

I have some questions on the actual effectiveness of the visual program set up. From my understanding, and from all the results shown in the paper, pySpatial lists out the procedures of calling external APIs. It does not do additional complex actions (e.g., loops, if/else, etc) beyond a sequence of API calling. Are there cases where the question answer requires more than a linear sequence of API calling and if so can we see several of these examples? If not, this makes me question whether explicitly converting it to program language code really helps with the set up or just a list of action items is fine.

In addition, one of the main differences between VADAR and pySpatial is that VADAR generates the APIs required for a particular task, while in your case a set of pre-defined APIs are set. I wonder if the set of API limits the type of spatial reasoning tasks you can perform. For example, I wonder if the model would still perform better on tasks such as absolute metrics such as distances and volumes estimation, where the reconstruction itself isn’t explicitly providing more information. It would also be great to see the results on more benchmarks, e.g., MMSI-Bench, All-Angle Bench, etc. It would also be great if you can highlight a bit more on the architectural difference of VADAR and pySpatial.

**Questions:**

Please refer to the weakness section for my main concerns. I would like to further understand the limitations of VADAR in terms of the complexity of the programs that it can generate, as well as more benchmarks to show the range of 3D spatial reasoning tasks it can perform.

---

> ### Author Response · Authors · 2025-11-22
> **Response to Reviewer Rgza (Part 1/2)**
>
> Dear Reviewer Rgza,
>
> Thank you for your insightful comments. We provide point-by-point responses to address your concerns below.
>
> ---
>
> **Comment (1)**: “*Are there cases where the question answer requires more than a linear sequence of API calling and if so can we see several of these examples? If not, this makes me question whether explicitly converting it to program language code really helps with the set up or just a list of action items is fine.*”
>
> **Response (1)**: Thank you for your insightful comments, and we would like to clarify this misunderstanding. In the instructions and in-context learning examples provided to the code agent, we explicitly encourage the generation of plain visual program code in order to minimize potential errors during program execution. Moreover, most queries in the MindCube benchmark can indeed be solved with relatively “linear” programs. That said, pySpatial is fully capable of producing more complex code that incorporates control-flow structures such as `for` loops and `if` statements if needed. We present two examples in Appendix A.3 of the revised paper. As shown, our pySpatial framework can execute control-flow operations such as for-loops and functional compositions, enabling it to build complex 3D operations and reason robustly across diverse spatial reasoning tasks.
>
> ---
>
> **Comment (2)**: “*In addition, one of the main differences between VADAR and pySpatial is that VADAR generates the APIs required for a particular task, while in your case a set of pre-defined APIs are set. I wonder if the set of API limits the type of spatial reasoning tasks you can perform. For example, I wonder if the model would still perform better on tasks such as absolute metrics such as distances and volumes estimation, where the reconstruction itself isn’t explicitly providing more information.*”
>
> **Response (2)**: We acknowledge that the use of a pre-defined API may limit the range of tasks our approach can currently support. However, we would like to clarify that this limitation stems primarily from the capabilities of the underlying base models (or the vision specialist module, as noted in the VADAR paper), rather than from the API design itself. The same constraint also applies to VADAR: despite its adaptive API design, it cannot perform depth-related tasks without access to a depth estimation model. Therefore, we believe this limitation is inherent to all visual programming approaches. On the other hand, this limitation can naturally be mitigated through engineering efforts by incorporating additional vision specialist modules, which would enable the system to support a broader range of tasks.
>
> We would also like to clarify that our framework can handle tasks involving absolute metrics, such as distance estimation, since recent 3D reconstruction models like CUT3R can indeed provide this information. This capability is further demonstrated in our real-world experiments (Figure 3), where our framework outputs accurate distances and turn angles that enable a quadrupedal robot to successfully navigate to the target destination.

---

> ### Author Response · Authors · 2025-11-22
> **Response to Reviewer Rgza (Part 2/2)**
>
> **Response (3)**: Below we provide a more detailed comparison between VADAR and our pySpatial framework:
> - **VADAR**: VADAR adopts a two-stage program synthesis pipeline. In the first stage, it composes basic functions and APIs; in the second stage, it generates the full program. The system includes multiple agents, including a signature agent, an implementation agent, and a program agent, each guided by different sets of manually crafted prompts. In terms of tools, VADAR primarily relies on 2D vision modules, such as segmentation and monocular depth estimation.
> - **Our pySpatial**: In contrast, pySpatial follows a simpler and more direct design. We define a unified 3D API manually, and a single code agent is responsible for generating the entire visual program using this API. More importantly, pySpatial integrates 3D-capable modules, including 3D reconstruction, natural-language-based movement descriptions, and novel-view synthesis, enabling the system to perform geometry-aware reasoning that goes beyond VADAR’s 2D toolset.
>
> We would like to point out that although VADAR can generate more complex code due to its adaptive API design, it has two clear disadvantages compared with our approach:
> - It can only perform spatial reasoning on single-view images, where the range of spatial reasoning tasks is inherently limited and considerably simpler.
> - The two-stage pipeline of VADAR (signature/API generation followed by program synthesis) produces more complex program structures but does not necessarily lead to better task performance. In fact, as shown in Table 2, when we equip VADAR with our 3D reconstruction tools to enable fair comparison, its performance actually declines relative to the original VADAR setup. This suggests that the additional complexity introduced by its multi-agent, multi-stage design does not translate into improved spatial reasoning ability and may even introduce instability in program generation.
>
> Overall, these differences highlight that pySpatial offers a more streamlined architecture with stronger practical effectiveness in multi-view spatial reasoning tasks.
>
> ---
>
> **Comment (4)**: “*It would also be great to see the results on more benchmarks, e.g., MMSI-Bench, All-Angle Bench, etc.*”
>
> **Response (4)**: Thank you for the suggestion. We were not aware of these concurrent papers at the time of submission, and thanks for bringing them to our attention. Following your comments, we have evaluated our method on MMSI-Bench, and the corresponding results are reported below.
>
> | Method | Cam.--Cam. | Obj.--Obj. | Reg.--Reg. | Cam.--Obj. | Obj.--Reg. | Cam.--Reg. | Meas. | Appr. | Cam. | Obj. | MSR | Avg. |
> |--------|------------|------------|------------|------------|------------|------------|-------|-------|------|------|-----|------|
> | GPT-4.1 | 36.6 | 26.6 | 27.2 | 29.1 | 36.5 | 27.7 | 37.5 | 24.2 | 36.5 | 32.9 | 28.8 | 30.9 |
> | + pySpatial | 51.6 | 28.7 | 27.2 | 20.9 | 41.2 | 38.6 | 46.9 | 39.4 | 46.0 | 38.2 | 36.4 | 37.3 |
>
> We observe that pySpatial improves the overall MMSI-Bench performance by 6.4% on average, further demonstrating the general effectiveness of our approach. We have included these results into Appendix A.1 of the revised paper.
>
> ---
>
> We hope our responses have addressed your concerns. If you have additional comments or concerns, please let us know and we will be more than happy to answer.
>
> Best,
>
> Authors

---

### Official Review · Reviewer_ZMiG · 2025-11-02

**Soundness:** 3
**Presentation:** 4
**Contribution:** 3
**Rating:** 8
**Confidence:** 4

**Summary:**

This paper presents pySpatial, a visual programming framework that enables MLLMs to perform explicit 3D spatial reasoning. The core contribution is a Python-based API that allows MLLMs to compose spatial tools—including feed-forward 3D reconstruction, camera pose estimation, and novel view synthesis—through code generation. Given sparse image views and natural language queries, the system generates executable programs that transform 2D observations into explorable 3D representations, enabling geometric reasoning rather than relying on implicit mental models.
The framework operates in a zero-shot setting without gradient-based fine-tuning. Experimental results demonstrate substantial improvements over MLLM baselines on MindCube (+12.94% over GPT-4.1-mini) and Omni3D-Bench benchmarks. The authors also provide qualitative validation through real-world robot navigation experiments. The interpretability of generated programs and modular tool composition are key practical advantages.

**Strengths:**

1. Novel and Well-Motivated Problem Formulation

The paper addresses a clearly identified limitation in current MLLMs regarding 3D spatial reasoning from limited views (Section 1, lines 54-65). While recent works like SpatialVLM and SpatialRGPT focus on single-view spatial understanding, this work tackles the more challenging multi-view setting where models must reason across perspectives. The visual programming paradigm is well-suited to this problem, allowing flexible composition of spatial tools through a clean Python API (Code 1, Section 3.2).

2. Strong and Consistent Empirical Results Across Multiple Benchmarks

On the challenging MindCube benchmark (Table 1), pySpatial achieves 58.56% overall accuracy, substantially outperforming the best open-weight model DeepSeek-VL2-Small by 10.94% and the strongest proprietary baseline GPT-4.1-mini by 12.94%. The results are particularly impressive on the "Among" category (60.54%), where all baseline approaches fail to exceed 50%—this category requires reasoning about how a central object relates to all surrounding objects, a challenging spatial task. Notably, pySpatial outperforms VLM-3R by 16.5% while operating entirely zero-shot. On MindCube-1k (Table 2), pySpatial with GPT-4o reaches 62.67%, outperforming spatial mental models by ~20%. The framework also generalizes to single-view settings, achieving state-of-the-art on Omni3D-Bench among visual programming approaches (+3.8% over VADAR, Table 3) and even surpassing GPT-4o on total score.

3. Zero-Shot Operation with Broad Generalization

A key practical advantage is that pySpatial requires no gradient-based fine-tuning (lines 47-48, 106-107). The ablation study (Table 4, Section 4.5) demonstrates that augmenting different MLLMs with pySpatial consistently yields substantial improvements: GPT-4o improves from 42.29% to 62.67%, GPT-4.1-mini from 43.34% to 58.19%, and GPT-4.1 from 44.67% to 63.42% on MindCube-1k. This plug-and-play nature makes the framework broadly applicable without requiring expensive task-specific training, and the consistent gains across different code agents validate the robustness of the approach. The framework also successfully transfers from multi-view (MindCube) to single-view (Omni3D-Bench) settings, demonstrating versatility.

4. Interpretability, Transparency, and Reproducibility

Unlike black-box MLLM approaches, pySpatial generates executable Python programs (Figure 2) that are human-readable and include well-structured comments explaining the reasoning process (e.g., "# Step 2: rotate the camera to the right from viewpoint 2 to see what is on the right side of the black chair"). This interpretability allows researchers to inspect, debug, and modify generated programs. The API is thoroughly documented (Code 1 in main paper, full specification in Appendix A Code 2), and prompts are provided in Appendix B (lines 892-1079). The modular design separates high-level reasoning from low-level execution (lines 220-223), and the paper commits to releasing code upon acceptance (line 266). Implementation details are comprehensive, including specific model choices (VGGT, CUT3R), rendering backend (Open3D), and computational setup (single A6000 GPU, Section 4.1).


5. Comprehensive Experimental Evaluation and Analysis

The evaluation is thorough and multi-faceted: (1) Benchmark diversity: Tests on both multi-view (MindCube) and single-view (Omni3D-Bench) benchmarks; (2) Baseline breadth: Compares against open-weight MLLMs, proprietary models, specialized spatial models, and prior visual programming approaches (Tables 1-3); (3) Qualitative analysis: Figure 2 shows diverse examples with generated programs, 3D reconstructions, and outputs; (4) Ablation study: Table 4 validates effectiveness across different code agents; (5) Failure analysis: Manual examination of ~100 samples (Section 4.5, Figure 4) attributes errors to program generation (6%), 3D reconstruction (13%), and final reasoning (20%), providing transparency about system limitations; (6) Efficiency analysis: Reports 7.45s per query on MindCube-1k, competitive with VADAR's 17.25s (Section 4.5); (7) Real-world validation: Indoor robot navigation experiment (Figure 3, Section 4.4) demonstrates practical applicability.

**Weaknesses:**

1. Limited and Non-Rigorous Real-World Validation
While Section 4.4 presents robot navigation as evidence of practical effectiveness, the evaluation is limited: (a) Qualitative only: No quantitative success rates, path efficiency metrics, or safety margins are reported; (b) Single environment: Testing appears confined to one 50m² two-room laboratory; (c) Manual intervention: High-level position commands are "manually converted into temporal velocity targets" (lines 375-377), reducing the autonomy claim; (d) Limited comparison: Only one baseline (GPT-4.1) is tested, and failure modes are not systematically analyzed. The paper states the robot "successfully traverses complex environments" (line 46), but Figure 3 shows only one trajectory. To strengthen claims of "practical effectiveness," the authors should provide: success rates over multiple trials/environments, collision avoidance validation, automated planning-to-control integration, and discussion of failure recovery mechanisms.


2. Reconstruction Quality as Performance Bottleneck with Insufficient Analysis
The failure analysis (Section 4.5, Figure 4) reveals that 13% of errors stem from 3D reconstruction quality, and the paper acknowledges this dependency. However, the analysis lacks depth: (a) No sensitivity studies: How does performance vary with 2 vs 4 vs 8 input views? What about different view overlap percentages or baseline distances? (b) Scene characteristics unexplored: Which scene properties (low texture, reflective surfaces, repetitive patterns, dynamic objects) most affect reconstruction quality? (c) Backend comparison incomplete: VGGT is used for most experiments, CUT3R for navigation, but no systematic comparison between VGGT/DUSt3R/CUT3R on the same multi-view tasks; (d) Mitigation strategies absent: Are there ways to detect reconstruction failures (e.g., uncertainty estimation) or fallback mechanisms when reconstruction quality is poor? The statement "advances in 3D reconstruction... hold the potential to further enhance our performance" (lines 469-470) is true but sidesteps the question of current robustness limits.


3. Heavy Reliance on Proprietary Models Limits Reproducibility
Main results depend critically on GPT-4o/GPT-4.1-mini (Table 1 uses GPT-4.1-mini "due to budget constraints," line 273; Table 2 and most analyses use GPT-4o). While Table 4 shows the framework works with different code agents, all are GPT-4 series models. The paper lacks: (a) Open-source code generation results: No experiments with CodeLlama, DeepSeek-Coder, or other fully open models for program synthesis; (b) Reproducibility concerns: Proprietary model APIs can change behavior across versions, making exact reproduction difficult; (c) Cost analysis: No discussion of API costs versus open alternatives; (d) Sensitivity to model choice: Beyond the three GPT-4 variants shown, how much does performance degrade with weaker but open code generators? This dependency limits the "zero-shot" and "plug-and-play" accessibility claims for researchers without API access or budgets.


4. Insufficient Statistical Rigor and Evaluation Details
The quantitative results lack statistical validation: (a) No confidence intervals: Tables 1-3 report point estimates without error bars or standard deviations; (b) No significance testing: Claims of improvement (e.g., "+12.94% over GPT-4.1-mini") are not tested for statistical significance; (c) Limited failure analysis scale: Only "about 100 samples" manually examined (line 460), representing <1% of MindCube and <10% of MindCube-1k; (d) Baseline parity unclear: No discussion of whether baselines use identical prompts, temperature settings, or number of trials; (e) Prompt sensitivity unstudied: The approach relies heavily on in-context examples (Appendix B), but no ablation on prompt variations or example selection is provided. Given that results are based on sampling from language models (which can be stochastic), statistical rigor is essential for verifying claimed improvements are robust rather than within noise margins.


5. Underspecified Calibration and Scale Handling
Critical details about camera calibration and scale are missing or unclear: (a) VGGT scale ambiguity: VGGT produces "up-to-scale" reconstructions (line 182), but Section 3.2's camera description assumes computing actual displacements (equations 1, line 191). How is scale determined or normalized for motion categorization? (b) Intrinsics estimation: Camera intrinsics K appear in Equation 1, but it's unclear whether these are estimated by the reconstruction model, extracted from metadata, or assumed as default values; (c) Coordinate frame alignment: How are multiple views aligned into a consistent world frame, especially for scenes without significant overlap? These technical details are important for understanding when the approach will succeed or fail and for enabling reproduction.

**Questions:**

1. Your main results use GPT-4o/GPT-4.1-mini. Can you report results using fully open-source code generation models (e.g., CodeLlama-34B, DeepSeek-Coder-33B, or recent Qwen-Coder models) for both program synthesis and final answer generation? This would: (a) Improve reproducibility for researchers without API access; (b) Clarify how much performance depends on proprietary model capabilities; (c) Identify minimum model requirements for the framework to be effective. Even negative results would be valuable for the community.

2. Please clarify the technical pipeline for camera parameters: (a) For VGGT (up-to-scale), how do you handle scale ambiguity in Equation 1 and the camera motion description (Section 3.2)—is scale normalized per scene, or are only relative directions used? (b) How are camera intrinsics K determined—estimated by reconstruction models, from metadata, or assumed? (c) For navigation with CUT3R, what is the sensitivity to metric scale errors (e.g., ±10-20%)? (d) How are camera extrinsics aligned to a consistent world frame for scenes with limited overlap? This is critical for reproducibility.

3. Can you provide: (a) Confidence intervals or bootstrapped standard errors for the main benchmark results (Tables 1-3)? (b) Statistical significance tests comparing pySpatial against key baselines (e.g., GPT-4.1-mini, VADAR)? (c) Ablation on prompt design—how sensitive is performance to the choice of in-context examples, API documentation verbosity, or structured output formatting? (d) Results with multiple random seeds or temperature settings to quantify variance? This would increase confidence that reported improvements are robust.

4. Given that 13% of failures stem from reconstruction quality (Section 4.5), can you provide systematic analysis of performance under varying input conditions? Specifically: (a) Performance curves as a function of number of input views (e.g., 2, 4, 6, 8 views); (b) Impact of view overlap percentage on reconstruction and final task accuracy; (c) Comparison between VGGT, DUSt3R, and CUT3R on the same MindCube subset; (d) Scene characteristics that most correlate with reconstruction failures (texture, lighting, clutter level)? This would help users understand when pySpatial is reliable.

---

> ### Author Response · Authors · 2025-11-22
> **Response to Reviewer ZMiG (Part 1/2)**
>
> Dear Reviewer ZMiG,
>
> Thank you for your insightful comments and positive recommendation of our work. We provide point-by-point responses to address your concerns below.
>
> ---
>
>
> **Comment (1)**: “*Limited and Non-Rigorous Real-World Validation…*”
>
> **Response (1)**: Thank you for your thoughtful feedback. Following your comments, we extended our real-world experiments by incorporating 10 additional scenarios that cover more diverse spatial layouts and object configurations. These added evaluations allow us to better assess the robustness of our approach under varying real-world conditions, and the results, as shown below, further support the generalization capability of our 3D reasoning framework.
>
> | Method      | GPT-4o | GPT-5 | pySpatial |
> |-------|---------|----------|----------|
> | Success Rate |  1 / 10 | 2 / 10  | 8 / 10  |
>
> We observe that both GPT-4o and GPT-5 are only able to handle relatively simple navigation scenarios, typically those involving two views with minimal viewpoint change and no required turns. In more challenging tasks, such as the multi-view indoor navigation example shown in Figure 3, GPT-series models consistently fail, often colliding at the very first corner of the expected trajectory. In contrast, pySpatial successfully completes these tasks by leveraging accurate 3D reconstruction and geometry-aware reasoning. In the most difficult cases, such as long hallways with repeated structural patterns or complex outdoor scenes with large open spaces, pySpatial also struggles and may fail to reach the goal. We will update our project website to include the corresponding video demonstrations.
>
> We have also included an additional qualitative example in a challenging dynamic scene to Appendix A.4 of the revised paper, where the views are captured at different times and a person is moving through the environment. In this setting, pySpatial remains effective since the reconstruction module is able to handle the dynamic elements, while GPT-5 continues to fail to generate a safe or correct navigation trajectory. These extended real-world experiments further validate the robustness and practical effectiveness of our approach.
>
> ---
>
> **Comment (2)**: “*Reconstruction Quality as Performance Bottleneck with Insufficient Analysis…*”
>
> **Response (2)**: Thanks for your comments and we have conducted the following studies to address your concerns:
> - We performed a sensitivity analysis on the number of views using MindCube-1k. The results show that performance improves smoothly as more views are provided through linear interpolation, but the gains begin to saturate after a modest number of inputs.
> - We performed a comparison with different backends, including VGGT, Pi3, and CUT3R, under the same multi-view input setting. The results are as follows.
>
>    | Method      | Overall | rotation | Among | around |
>    |------------------|---------|----------|-------|--------|
>    | GPT-4o        | 42.29   | 35.00    | 43.00 | 46.20  |
>    | pySpatial w/ VGGT   | 62.67   | 41.00    | 66.33 | 71.20  |
>    | pySpatial w/ Pi3    | 63.33   | 43.50    | 65.66 | 72.33  |
>    | pySpatial w/ CUT3R  | 61.05   | 40.50    | 64.33 | 69.60  |
>
>    The results show that pySpatial achieves stable performance across different reconstruction backends, confirming that our method is robust with backbone selection.
> - We included an initial study on mitigation reconstruction failures via an oracle setting. Specifically, we manually identify reconstruction failures and apply a simple fallback mechanism: when the multi-view reconstruction is unreliable, the system bypasses pySpatial and defaults to standard MLLM inference. This strategy successfully prevents catastrophic reconstruction errors; however, overall performance still declines slightly, as expected, since fallback removes access to the spatial cues that are crucial for fine-grained 3D reasoning. We will continue to explore this direction, and thank you for suggesting this interesting idea.

---

> > ### Comment · Reviewer_ZMiG · 2025-11-25
> > **rebuttal addressed my concerns**
> >
> > The authors have substantially addressed my concerns through additional experiments.
> >
> > Key improvements include: (1) demonstration of compatibility with open-source code generation models (e.g. Qwen3-Coder, DeepSeek-V3), significantly improving accessibility and reproducibility; (2) quantitative real-world validation across 10 navigation scenarios (8/10 success rate vs 1/10 for GPT-4o); (3) backend robustness analysis across multiple reconstruction methods.
> >
> > These additions strengthen the empirical validation and practical impact of the work.

---

> > > ### Author Response · Authors · 2025-11-27
> > > **Thank you very much for recognizing our improvements**
> > >
> > > We are glad that our rebuttal helped address your earlier concerns. Your feedback has been truly helpful in strengthening our work, and we are sincerely grateful for your time and constructive feedback!

---

> ### Author Response · Authors · 2025-11-22
> **Response to Reviewer ZMiG (Part 2/2)**
>
> **Comment (3)**: “*Heavy Reliance on Proprietary Models Limits Reproducibility Main results depend critically on GPT-4o/GPT-4.1-mini…*”
>
> **Response (3)**: Thanks for pointing this out. In addition to the proprietary models reported in the main paper, we have now included experiments with fully open-source code generation models. Specifically, we evaluate our pySpatial with Qwen-Coder and DeepSeek-V3 under the same program synthesis setting. Both models demonstrate strong compatibility with our pySpatial pipeline and follow the same trend as the larger proprietary models, further supporting the generality of our approach.
>
> | Method               | Overall | Rotation | Among | Around |
> |----------------------|---------|----------|--------|--------|
> | GPT-4o           | 42.29   | 35.00    | 43.00 | 46.20  |
> | pySpatial w/ GPT-4o      | 62.67   | 41.00    | 66.33 | 71.20
> | pySpatial w/ Qwen3-Coder| 62.10   | 40.00    | 64.50 | 74.00  |
> | pySpatial w/ DeepSeek-v3| 61.05   | 40.50    | 65.33 | 64.80  |
>
> We have added these results and corresponding analysis to Appendix A.2 of the revised paper.
>
>
> ---
>
>
>
> **Comment (4)**: “*Insufficient Statistical Rigor and Evaluation Details…*”
>
> **Response (4)**: Here are the detailed responses:
>
> - We have now added confidence intervals for our main results on the MindCube-1k benchmarks. Specifically, we report mean performance along with 95% confidence intervals computed over three random seeds. These intervals have been included in Table 2 of the revised manuscript to provide a clearer picture of the statistical reliability of our results. We will also update the results on the full MindCube benchmark accordingly.
>
> - We clarify that all baselines are evaluated under identical generation configurations (including temperature, top-p, and max token settings). As for the prompts, our pySpatial pipeline introduces additional task-specific guidance to facilitate reliable program generation and spatial reasoning. For reproducibility, we have clearly documented all prompts used for our method in the appendix. We will also open-source the code upon acceptance of this work to facilitate reproducibility.
>
> - Further, we conducted an ablation study that examines the effect of varying the number of in-context learning examples in the code generation prompts on MindCube-1k.
>
>    | Method      | Overall | rotation | Among | around |
>    |------------------|---------|----------|-------|--------|
>    | GPT-4o           | 42.29   | 35.00    | 43.00 | 46.20  |
>    | pySpatial w/ 0 examples | 53.62   | 32.50   | 55.16 | 66.80  |
>    | pySpatial w/ 2 examples | 62.67   | 41.00  | 66.33 | 71.20  |
>    | pySpatial w/ 4 examples |  63.14 | 46.00 |   64.50     |    73.60       |
>
>    These results suggest that pySpatial benefits from additional examples, but even minimal in-context supervision is sufficient to unlock strong spatial reasoning capabilities. We have also added these results and corresponding analysis to Appendix A.2.
>
>
> ---
>
> **Comment (5)**: “*Underspecified Calibration and Scale Handling…*”
>
> **Response (5)**: Thanks for your careful review of our paper. Here are the detailed responses:
> - VGGT predicts point clouds in a normalized unit space. We empirically set the moving distance to 0.3. Also we’d like to point out that the model is able to change the distance because it is a changeable parameter for both moving functions. We have now clarified this in Lines 223-225.
> - The camera intrinsics is assumed to be the same across all frames and is estimated from the reconstruction model. We have clarified this point in Lines 162-165.
> - We argue that coordinate frame alignment is a functionality that reconstruction models, e.g. VGGT,  provide. Specifically, VGGT is trained for multi-view matching on large scale data, therefore it is able to handle scenes without significant overlap. We also clarified this point in Line 136. Please let us know if you find it unclear.
>
> ---
>
> We sincerely appreciate your thoughtful comments, which have helped us improve the paper. If you have any further questions or concerns, please don’t hesitate to let us know.
>
> Best,
>
> Authors

---

### Author Response · Authors · 2025-11-22
**General Response**

Dear AC and Reviewers,

We are sincerely grateful to you all for dedicating time and efforts in providing these detailed and thoughtful reviews, which helped us to improve the quality of our paper. We have also carefully revised the paper based on your thoughtful feedback. For your convenience, we have highlighted all the revisions made compared to the initial version in blue.

Here, apart from the point-by-point responses to each reviewer, we would like to summarize the contributions of this work and highlight our new results added during the rebuttal phase.

---

We are delighted that the reviewers appreciate and recognize the following strengths and contributions:
- The paper addresses a clearly identified limitation in current MLLMs regarding 3D spatial reasoning from limited views. The motivation is reasonable and reflects an active research direction in improving spatial understanding for embodied and multi-view settings. The problem of spatial reasoning with MLLMs is also an important and relevant task in the community. **[All Reviewers]**
- The proposed visual programming framework is well-structured and methodologically sound. The method is well described. Unlike black-box MLLM approaches, pySpatial generates executable Python programs that are human-readable and include well-structured comments explaining the reasoning process. **[ZMiG, Rgza, Xvqg]**
- The experimental evaluation is thorough. The framework demonstrates consistent zero-shot performance gains on both multi-view (MindCube) and single-view (Omni3D-Bench) benchmarks. **[All Reviewers]**

---

In the discussion phase, we have included the following key experiments and clarifications:
- We additionally report results on MMSI-Bench, where pySpatial improves the average performance by 6.4% over the GPT-4o baseline, demonstrating the general effectiveness of our approach across a broad range of spatial reasoning tasks. The full results are provided in Appendix A.1 of the revised paper.
- We evaluated pySpatial using open-source coding models (Qwen3-Coder and DeepSeek-V3) and observed performance comparable to that achieved with GPT-4o. This finding indicates that pySpatial is not reliant on proprietary code agents and can be effectively paired with widely accessible open-source alternatives. The results are incorporated into Appendix A.2 of the revised paper.
- We also evaluated pySpatial using different 3D reconstruction backbones, including VGGT, Pi3, CUT3R, and observed substantial improvements over the GPT-4o baseline across all settings. This indicates that our framework is robust to the choice of reconstruction method. We have added these results to Appendix A.2 of the revised paper.
- We further analyzed how the number of in-context learning examples provided to the code agent influences the performance of pySpatial. The results show that while pySpatial does benefit from additional examples, even minimal in-context supervision is sufficient to unlock strong spatial reasoning capabilities of our framework. The results and discussions are also incorporated into Appendix A.2 of the revised paper.
- We clarified that our pySpatial is also capable of generating visual programs with more expressive control-flow constructs, including for-loops and functional compositions, allowing it to construct complex multi-step 3D operations. We presented two additional qualitative examples in Appendix A.3 of the revised paper.
- We added 10 new real-world navigation scenarios spanning diverse spatial layouts and dynamic environments. pySpatial succeeds in 8 out of 10 cases, substantially outperforming both GPT-4o and GPT-5. Additional dynamic-scene examples are provided in Appendix A.4, further demonstrating that our framework remains robust even in the presence of dynamic objects within the scene.

---

We greatly appreciate your time and constructive feedback on our work. If you have remaining questions or concerns, please do not hesitate to let us know and we will be happy to address them.


Best,

Authors

---

### Meta-Review · Area_Chair_xL4U · 2025-12-30

**Summary:**

The paper presents pySpatial, a visual programming framework that composes spatial tools to enable MLLMs to perform explicit 3D spatial reasoning.

The initial reviewer scores were 8, 4, 6, and 4. All reviewers agreed that the paper is clearly written and demonstrates strong empirical performance. The primary concerns raised include:
 (1) missing additional ablation studies across different 3D reconstruction methods and LLMs (ZMiG, FeNP, Xvqg),
 (2) the limited scale of real-world experiments (ZMiG, Xvqg),
 (3) limited novelty relative to prior work on visual programming (Rgza, Xvqg), and
 (4) the lack of evaluation on a broader set of benchmarks (Rgza).

In the rebuttal, the authors provided additional experimental results, which adequately addressed most of the reviewers’ concerns.

While the conceptual framework itself is incremental, the AC believes that the paper constitutes a solid empirical contribution. The results are thorough and offer useful insights for the community working on spatial reasoning and tool-based MLLM systems.

Considering the clarity of presentation and empirical strength, the AC recommends accepting the paper. The authors should revise the paper according to reviewers’ comments.

**Reviewer Concerns:**

See above.

**Reviewer Scores:**

The reviewers are likely to increase their scores.

---

### Decision · Program_Chairs · 2026-01-26

Accept (Poster)